# SCAND1 Reverses Epithelial-to-Mesenchymal Transition (EMT) and Suppresses Prostate Cancer Growth and Migration

**DOI:** 10.3390/cells11243993

**Published:** 2022-12-10

**Authors:** Takanori Eguchi, Eva Csizmadia, Hotaka Kawai, Mona Sheta, Kunihiro Yoshida, Thomas L. Prince, Barbara Wegiel, Stuart K. Calderwood

**Affiliations:** 1Department of Dental Pharmacology, Graduate School of Medicine, Dentistry and Pharmaceutical Sciences, Okayama University, Okayama 700-8525, Japan; 2Division of Surgical Sciences, Department of Surgery, Cancer Research Institute, Beth Israel Deaconess Medical Center, Harvard Medical School, Boston, MA 02115, USA; 3Department of Oral Pathology and Medicine, Graduate School of Medicine, Dentistry and Pharmaceutical Sciences, Okayama University, Okayama 700-8525, Japan; 4Department of Cancer Biology, National Cancer Institute, Cairo University, Cairo 11796, Egypt; 5Department of Oral and Craniofacial Surgery, Graduate School of Medicine, Dentistry and Pharmaceutical Sciences, Okayama University, Okayama 700-8525, Japan; 6Ranok Therapeutics, Waltham, MA 02451, USA; 7Department of Radiation Oncology, Beth Israel Deaconess Medical Center, Harvard Medical School, Boston, MA 02115, USA

**Keywords:** epithelial-to-mesenchymal transition (EMT), hybrid E/M, partial EMT, SCAND1, MZF1, SCAN zinc finger transcription factors, gene expression, cancer prognosis, collective migration, metastasis

## Abstract

Epithelial–mesenchymal transition (EMT) is a reversible cellular program that transiently places epithelial (E) cells into pseudo-mesenchymal (M) cell states. The malignant progression and resistance of many carcinomas depend on EMT activation, partial EMT, or hybrid E/M status in neoplastic cells. EMT is activated by tumor microenvironmental TGFβ signal and EMT-inducing transcription factors, such as ZEB1/2, in tumor cells. However, reverse EMT factors are less studied. We demonstrate that prostate epithelial transcription factor SCAND1 can reverse the cancer cell mesenchymal and hybrid E/M phenotypes to a more epithelial, less invasive status and inhibit their proliferation and migration in DU-145 prostate cancer cells. SCAND1 is a SCAN domain-containing protein and hetero-oligomerizes with SCAN-zinc finger transcription factors, such as MZF1, for accessing DNA and the transcriptional co-repression of target genes. We found that SCAND1 expression correlated with maintaining epithelial features, whereas the loss of SCAND1 was associated with mesenchymal phenotypes of tumor cells. SCAND1 and MZF1 were mutually inducible and coordinately included in chromatin with hetero-chromatin protein HP1γ. The overexpression of SCAND1 reversed hybrid E/M status into an epithelial phenotype with E-cadherin and β-catenin relocation. Consistently, the co-expression analysis in TCGA PanCancer Atlas revealed that SCAND1 and MZF1 expression was negatively correlated with EMT driver genes, including CTNNB1, ZEB1, ZEB2 and TGFBRs, in prostate adenocarcinoma specimens. In addition, SCAND1 overexpression suppressed tumor cell proliferation by reducing the MAP3K-MEK-ERK signaling pathway. Of note, in a mouse tumor xenograft model, SCAND1 overexpression significantly reduced Ki-67(+) and Vimentin(+) tumor cells and inhibited migration and lymph node metastasis of prostate cancer. Kaplan–Meier analysis showed high expression of SCAND1 and MZF1 to correlate with better prognoses in pancreatic cancer and head and neck cancers, although with poorer prognosis in kidney cancer. Overall, these data suggest that SCAND1 induces expression and coordinated heterochromatin-binding of MZF1 to reverse the hybrid E/M status into an epithelial phenotype and, inhibits tumor cell proliferation, migration, and metastasis, potentially by repressing the gene expression of EMT drivers and the MAP3K-MEK-ERK signaling pathway.

## 1. Introduction

Epithelial–mesenchymal transition (EMT) is a reversible cellular program that transiently transforms epithelial (E) cells into pseudo-mesenchymal (M) cell states [1,2,3,4,5,6]. During this process, epithelial cells progressively lose their cobblestone appearance observed in monolayer cultures to adopt a spindle-shaped mesenchymal morphology. EMT plays a significant role in the malignant progression of many types of carcinomas [7,8,9,10]. During the tumor progression, the EMT program confers multiple traits associated with high-grade malignancy on carcinoma cells [7,11,12]. Normally, the cells forming epithelial sheets in various adult tissues display apical–basal polarity and are held together laterally by the adherens junction and tight junction. Adherens junctions are formed by cell surface epithelial cadherin (E-cadherin) molecules encoded by the *ECAD* gene, while tight junctions are formed by tight junction proteins TJP (also known as *zonula occludens*: ZO) and epithelial cell adhesion molecule EpCAM (also known as CD326) [10,13]. This organization is essential for the structural integrity of epithelia. Upon the activation of EMT, E-cadherin expression is repressed, which leads to the loss of the typical polygonal, cobblestone morphology of epithelial cells [10,12]. Cells acquire a spindle-shaped mesenchymal morphology and express markers, including vimentin (encoded by the *VIM* gene), neural cadherin (N-cadherin), and fibronectin [3]. Recent studies have shown that EMT is rarely executed as an on/off phenomenon in cancer, as the process is rather gradual and often remains incomplete, termed partial EMT, hybrid EMT, or hybrid E/M [2,14]. Hybrid EMT was identified by the co-expression of epithelial (EpCAM+) and mesenchymal (Vim+) marker genes in an autochthonous murine prostate cancer model [15]. Consistently, hybrid EMT of prostate cancer cells within a tumor organoid (tumoroid) was identified by the co-expression of epithelial (E-cadherin+ EpCAM+) and mesenchymal (Vimentin+) markers with enhanced stemness [13]. Another group identified different tumor transition states during EMT in cancer progression and introduced the term hybrid EMT [16,17]. The dynamic induction of EMT and MET changes the cellular phenotypes of carcinoma cells. Drug sensitivity, proliferation, and response to apoptosis signals are highest in more epithelial states, whereas drug efflux, invasion, and immune evasion are highest in more mesenchymal states [18,19]. A hybrid EMT state provides maximal stemness, tumor initiation capacity, and the ability to adapt to environmental changes [18].

The process of EMT is orchestrated by EMT-inducing transcription factors (EMT-TFs), which act combinatorially to induce the expression of genes that promote the mesenchymal cell state and repress the expression of genes that maintain the epithelial state [1,2,3,4,6]. These include the zinc-finger E-box binding homeobox factors ZEB1 and ZEB2, SNAIL (also known as SNAI1), SLUG (also known as SNAI2), and the basic helix–loop–helix factors TWIST1 and TWIST2. These EMT-TFs regulate the expression of each other and, in different combinations, induce the expression of hundreds of genes associated with the mesenchymal state (vimentin, N-cadherin, fibronectin, integrin β1 and β3, and metalloproteinases), while repressing genes associated with the epithelial state (E-cadherin, EpCAM, occludins, claudins, and cytokeratins). However, less is known about anti-EMT transcription factors (TF) that reverse EMT programs and place pseudo-mesenchymal cancer cells back into an epithelial phenotype. Here, we aim to address such factors.

The SREZBP-CTfin51-AW1-Number 18 cDNA (SCAN) domain is a leucine-rich oligomerization domain, highly conserved among the SCAN domain-containing transcription factor (SCAN-TF) family. This family contains 64 members in humans, most of which contain a zinc finger (ZF) domain, hence SCAN-ZF factors [20,21,22,23,24]. Myeloid zinc finger 1 (MZF1), also known as ZSCAN6 or ZNF42, belongs to the SCAN-ZF family and contains an N-terminal SCAN domain, a linker region, and a C-terminal DNA binding domain [25,26,27]. Many studies have identified MZF1 as an oncogenic transcription factor [25,28,29,30,31] and cancer stemness factor [32,33]. However, MZF1 can also function as a tumor suppressor that, for instance, represses MMP2 in cervical cancer [34] and mediates oncogene-induced senescence [35]. Furthermore, MZF1 is also vital in a protumorigenic microenvironment by activating the osteopontin (*OPN*) gene in mesenchymal stem cells (MSC), leading to transformation into cancer-associated fibroblasts (CAF) [36] and the *VEGF* gene in tumor endothelial cells (TEC) [37]. 

Among 64 types of SCAN-TFs, 6 zinc fingerless SCAN domain-only proteins exist [21,22]. SCAND1 is a SCAN domain-only protein and hetero-oligomerizes with other SCAN-ZFs, including MZF1, through inter-SCAN domain interactions to repress transcription [23,24,28,38]. Hetero-oligomerization between SCAN domain-only molecules and SCAN-ZF molecules transforms their roles to form a transcriptional repressor complex [23,24,28,38]. Indeed, SCAND1 represses the *CDC37* gene (encoding cell division control 37) by interacting with MZF1 and suppressing prostate cancer progression [28]. However, the potential involvement of SCAND1 in EMT has not been addressed. Here, we investigate the involvement of SCAND1 in reversing EMT in cancer. 

## 2. Materials and Methods

### 2.1. Cell Culture

Prostate cancer cell lines PD-145 and PC-3 were provided by ATCC and cultured in DMEM and RPMI medium, respectively, with 10% FBS. The prostate normal epithelial cell line immortalized with SV40 large T antigen (PNT2) was purchased from Sigma (St. Louis, MO, USA) and cultured in RPMI medium with 10% FBS. Human normal prostate epithelial cells (PrEC) were purchased from Lonza (Basel, Switzerland) and cultured in prostate epithelial cell basal medium supplemented with bovine pituitary extract, hydrocortisone, hEGF, epinephrine, transferrin, insulin, retinoic acid, triiodothyronine and GA-1000 (Lonza).

To compare the E/M status of different cell types, we first seeded cells to be 10% confluent in each dish. We replaced media with fresh ones one day before taking pictures and sampling proteins and RNA, then corrected the samples at around 50% confluent status of cells.

### 2.2. cDNA Transfection and Stable Cell Cloning

We used pcDNA3/MZF1^Flag^ and pCMV6/SCAND1^myc-Flag^ as previously constructed [28]. The cDNA of SCAND1 variant 1 (accession number NM_016558) was purchased from OriGene (Rockville, MD, USA). DU-145 cells were transfected with these plasmids and pcDNA3 vector using Lipofectamine 2000 (Thermo Fisher Scientific, Waltham, MA, USA) and cultured with 0.2, 0.4, 0.8, and 1.6 μg/mL of geneticin for 2 weeks to establish stable clones. The surviving cells were used for subsequent assays. For overexpression of heterochromatin protein 1γ (HP1γ), also known as chromobox protein 3 (CBX3), we used pcDNA3.1/HP1γ as previously cloned [39,40].

### 2.3. Lentiviral Infection and Cell Cloning

Lentivirus infection was performed as previously described [12]. A lentiviral expression vector for SCAND1^HA3^ was constructed from pCMV6/SCAND1^myc-Flag^ [28] and pcDNA3.1-HA3 that contains triple hemagglutinin (HA) tag [40] via gene recombination. The pLV-SCAND1^HA3^ (4 μg), psPAX2 (3 μg), and pVSV-G (1 μg) were co-transfected using Lipofectamine LTX and Plus reagents (Thermo Fisher Scientific) into HEK293FT cells cultured in DMEM supplemented with 10% heat-inactivated (HI) FBS, penicillin and streptomycin. After 10 h, the medium was replaced with a fresh one. After 48 h, the cell culture supernatant was collected and centrifuged at 500× *g* for 10 min. The supernatant was filtered through Millex-HP 0.45 μm polyethersulfone low protein-binding filters (Millipore, Billerica, MA, USA). DU-145 cells were cultured in the half-and-half mixture of the conditioned medium containing pseudo-virus particles and DMEM with 10% FBS supplemented with 5 μg/mL of polybrene and centrifuged for spinfection at 800× *g* for 1 h at room temperature (RT). For selection, cells were cultured in DMEM with 10% HI-FBS and 1 μg/mL of puromycin for 10 days. The surviving cells were used for subsequent assays.

### 2.4. siRNA

siRNA was transfected as previously described [28]. MZF1 siRNA (SR305183, mixture of A, B, and C) and non-silencing siRNA (SR30004) were purchased from OriGene. Cells were seeded into 6-well plates at 200 thousand cells per well and transfected with siRNA (7.5 nM) using 7.5 µL Lipofectamine RNAiMax (Thermo Fisher Scientific) by a reverse-transfection method [41]. Total RNA was collected after 48 h post-transfection.

### 2.5. qRT-PCR

qRT-PCR was performed as previously described [40,42]. Total RNA was prepared with DNase I treatment using RNeasy (Qiagen, Hilden, Germany). cDNA was synthesized using QuantiTect kit (Qiagen), a mixture of oligo dT and random primers, then diluted 5-fold in 10 mM Tris-Cl and 0.1 mM EDTA buffer. A step dilution of the cDNA pool was prepared as a standard for relative expression. cDNA (4–10 µL), 0.25 µM of each primer, and 10 µL SYBR green 2× Master Mix (Applied Biosystems, Waltham, MA, USA) were mixed and filled up to a 20 µL of a reaction mixture. We designed and used primer pairs (Table 1). We validated the primer pairs by melting curve analysis of single amplicons, amplification efficiencies, band size in agarose gel electrophoresis, and DNA sequencing of PCR products. Initial denaturation was performed at 95 °C 10 min, and 40 cycles of PCR were run at 95 °C for 15 sec and at 60 °C for 1 min. Relative mRNA expression levels were obtained as compared with the standard described above.

### 2.6. Western Blotting

Protein sampling, SDS-PAGE, and Western blotting were performed as previously described [13,28,40]. Cells cultured in 60 mm dishes were washed with ice-cold PBS. Cells were soaked in 500 μL of ice-cold CelLytic M (Sigma) supplemented with a cocktail of protease inhibitors and phosphatase inhibitors (Thermo Fisher Scientific) and incubated for 15 min on a shaker. Cells were scraped from the bottom of the dish and collected into 1.5 mL tubes. The lysate was centrifuged at 12,000× *g* for 15 min, and the supernatant was collected as soluble (Sol.) cell lysate. The insoluble pellet was treated with SDS sample buffer containing β-mercaptoethanol, boiled at 95 °C for 10 min, and used as chromatin (Chr.) fractions. The protein concentration was measured using a BCA assay kit (Thermo Fisher Scientific). Equal amounts of the protein samples (10–30 μg) were mixed with 4× Laemmli sample buffer, boiled for 5 min at 95 °C, and then loaded to 12% polyacrylamide gels for SDS-PAGE. Proteins were transferred to PVDF membranes. Membranes were soaked in a blocking buffer containing 5% skim milk in TBS-T for 1 h, incubated with primary antibodies in the blocking buffer at 4 °C overnight, and then incubated with secondary antibodies in the blocking buffer at RT for 1 h. Membranes were washed thrice with TBS-T for 10 min at RT between the steps. Proteins were detected by chemiluminescence reactions. We used antibodies against SCAND1 (ab64828, Abcam, Cambridge, UK), MZF1 (C10502, Assay Biotechnology, Fremont, CA, USA), HP1γ (BMP0003, MBL Life Science, Tokyo, Japan), Flag-tag (M2, Sigma), and Vimentin (V9, MA1-06909, Thermo Fisher Scientific). Antibodies against E-cadherin (#4065), ZO-1 (#5406), phosphorylated ERK-1/2 Thr202/Thr204 (D13.14.4E, #4370P), total ERK-1/2 (137F5, #4695), phosphorylated NF-κB p65 S536 (93H1, #3033S), total NF-κB p65 (#4674S) and HSP90β (#5087) were purchased from Cell Signaling Technologies (CST, Danvers, MA, USA).

### 2.7. Post-translational modifications (PTM) and hotspot mutation

We used PhosphoSitePlus (phosphosite.org) v6.6.0.4 for the analysis of PTM, including phosphorylation, sumoylation, and ubiquitination. SUMO interaction motifs (SIM) and some PTMs were analyzed as previously described [25]. Hotspot mutations were analyzed using cBioPortal.

### 2.8. Immunocytochemistry and Confocal Microscopy

Immunocytochemistry and confocal laser scanning microscopy (CLSM) were performed as previously described [40,43]. Cells were cultured on 120 mm round coverslips coated with poly-D-Lysine/Laminin coat (BD Bioscience, Franklin Lakes, NJ, USA). Cells were fixed with 4% paraformaldehyde for 10 min and washed with PBS twice. Cells were permeabilized with 0.1% Triton X-100 for 10 min and washed with PBS twice. Cells were incubated in a blocking buffer containing 3% normal goat serum in PBS for 30 min, with primary antibodies at 4 °C overnight and then secondary antibodies at RT for 1 h in the blocking buffer. Cells were washed three times with PBS for 5 min between the steps. Cells were mounted within ProLong Gold Antifade Mountant (Thermo Fisher Scientific). Fluorescence images were acquired using Axio Vision CLSM (Zeiss, Oberkochen, Germany) with a camera AxioCam MR3 (Zeiss) and a filter set for DAPI, GFP, Cy3.5 (excitation wavelength: 580 nm) and Cy5 (excitation wavelength: 650 nm). We used combinations of anti-mouse (Ms) IgG and anti-rabbit (Rb) IgG against; HA-tag (16B12, Ms; Covance, Princeton, NJ, USA) vs. MZF1 (C10502, Assay Biotechnology, Rb) or E-cadherin (24E10, #4065, CST, Rb); Flag-tag (M2, Sigma, Ms) vs. HA-tag (#561, MBL, Rb); β-catenin (15B8, ab6301, Abcam, Ms) vs. E-cadherin (24E10, #4065, CST, Rb) or SCAND1 (ab64828, Abcam, Rb). Anti-mouse and anti-rabbit IgG conjugated with Alexa Fluor 350, 488, 594, 648 (Thermo Fisher Scientific, 1:1000) were used as secondary antibodies.

### 2.9. Xenograft

One million cells were subcutaneously injected into the flanks of nude mice (Charles River, Wilmington, MA, USA). After 2 months, mice were sacrificed, and tumors and pelvic lymph nodes were harvested. All surgeries were performed under isoflurane anesthesia, and efforts were made to minimize the suffering of mice. All the animals were housed under specific pathogen-free conditions. 

### 2.10. Immunohistochemistry (IHC)

Tissues were fixed in formalin for 48 h, dehydrated in a series of alcohol solutions with increasing percentages, treated with xylene, and then embedded in paraffin. Paraffin blocks were cut into 5 µm sections using a microtome and mounted on microscope glass slides. Sections were deparaffinized and rehydrated. Antigens were retrieved with 0.1 M sodium citrate pH 6.0 in a high-pressure cooker. For immunofluorescence, sections were incubated within 10% normal serum for 15 min, primary antibodies against SCAND1 (ab64828, Abcam, Rb) and E-cadherin (#14472, CST, Ms) overnight at 4 °C, secondary antibodies against rabbit IgG conjugated with Alexa Fluor 594 and mouse IgG conjugated with Alexa Fluor 488 at RT for 1 h, and Hoechst. Slides were washed three times with PBS between staining steps. Fluorescence images were acquired using Axio Vision CLSM (Zeiss). Alternatively, sections were incubated with primary antibodies against Vimentin (V9, MA1-06909, Thermo Fisher Scientific) or Ki-67 (DAKO) overnight at 4 °C followed by biotinylated secondary antibodies (Vector Laboratories, Newark, CA, USA) at RT for 1 h. Slides were washed twice for 5 min in PBS between the staining steps. Sections were incubated with avidin–biotin peroxidase complexes of the ABC kit (Vectastain, Vector Laboratories) at RT for 30 min and with 3, 3′-diaminobenzidine (ImmPACT™ DAB HRP Substrate, Vector Laboratories). Nuclei were counterstained with hematoxylin. Sections were covered with mounting medium and coverslips, then evaluated microscopically (Nikon, Minato City, Tokyo).

### 2.11. Tissue-Specific Gene Expression

For analysis of tissue-specific gene expression, we used BioGPS portal system (http://biogps.org).

### 2.12. Co-Expression Analysis

A data set of prostate adenocarcinomas (Project ID: TCGA-PRAD, dbGaP study accession: phs000178, PanCancer Atlas; *n* = 494 patients/samples) was analyzed with Spearman’s rank correlation coefficient of co-expression using cBioPortal [44,45]. The co-expression of SCAND1 and MZF1 was first examined. Then, the co-expression of SCAND1 and MZF1 versus genes encoding mitogen-activated protein kinases (including MAP3Ks, MAP2Ks, and MAPKs), TGFβ receptors (TGFBR1, TGFBR2, and TGFBR3), EMT-TFs (including ZEB1 and ZEB2) and β-catenin (CTNNB1) was analyzed. Genes of which the Spearman’s correlation versus both MZF1 and SCAND1 expression was less than −0.3 are depicted in graphs and listed in a table with *p*- and *q*-values.

### 2.13. Kaplan–Meier Analysis

Kaplan–Meier plotting from RNA-seq data was performed using KM plotter and GEPIA 2 [46,47]. We analyzed the overall survival of patients suffering from pancreatic ductal adenocarcinoma (DAC) *n* = 177, renal clear cell carcinoma (RCC) *n* = 530, head and neck squamous cell carcinoma (SCC) stage III (*n* = 78) using the KM plotter with auto select best cutoff, and prostate adenocarcinoma (AC) *n* = 639 using GEPIA 2.

### 2.14. Statistics

Values of the two groups were compared with an unpaired student’s *t*-test using GraphPad Prism 8 and Microsoft Excel. *p* < 0.05 was considered to indicate statistical significance. Data were expressed as mean ± SD unless otherwise specified.

## 3. Results

### 3.1. SCAND1 Expression Correlated with an Epithelial Phenotype

We first searched SCAND1 gene expression in various tissues using the BioGPS database. SCAND1 mRNA was highly expressed in normal prostate, liver, lung, thyroid and heart (Figure 1A). We next characterized prostate cancer cell lines and normal prostate-derived cells. It is known that prostate cancer cell line PC-3 is the most malignant, metastatic, castration-resistant prostate cancer cell line, while DU-145 is a moderately metastatic prostate adenocarcinoma cell line [13,28,48]. We experienced that both Vimentin and E-cadherin were detectable in PC-3 cells under several culture conditions, suggesting that prostate cancer cells could often have a E/M hybrid status [9,13]. We also used normal prostate cells and epithelial cell line PNT2 and first examined their morphologies in the monolayer culture. Prostate normal cells and PNT2 cell populations mostly appeared as epithelial cells with cobblestone morphology and intercellular adhesion and rarely contained spindle-shaped cells (Figure 1B). PC-3 and DU-145 cells were mixed populations of spindle-shaped mesenchymal cells and the remainder of the cobblestone morphology, a characteristic of epithelial cells, indicating that DU-145 and PC-3 had a hybrid E/M status. 

We also used normal prostate cells and epithelial cell line PNT2 and first examined their morphologies in the monolayer culture. Prostate normal cells and PNT2 cell populations mostly appeared as epithelial cells with cobblestone morphology and intercellular adhesion and rarely contained spindle-shaped cells (Figure 1B). PC-3 and DU-145 cells were mixed populations of spindle-shaped mesenchymal cells and the remainder of the cobblestone morphology, a characteristic of epithelial cells, indicating that DU-145 and PC-3 had a hybrid E/M status.

To confirm the link between morphology and gene expression, we next examined the differential gene expression of *ECAD* and *VIM* among these cell types. *ECAD* mRNA was highly expressed in normal prostate epithelial cells and PNT2 cells while being reduced in DU-145 and PC-3 cells (Figure 1C), consistent with their morphology. *VIM* mRNA was highly expressed in PC-3 and DU-145 but lost in PNT2 cells (Figure 1D), again consistent with their morphology. 

To assess the involvement role of SCAND1 and MZF1 in the E/M status, we performed Western blotting of MZF1, SCAND1, and epithelial and mesenchymal markers. SCAND1 was expressed in normal prostate cells, while its levels were reduced in prostate cancer PC-3 and DU-145 cells (Figure 1E), suggesting that SCAND1 expression declines along with prostate oncogenesis. MZF1 was well-expressed in epithelial cell line PNT2, although MZF1 was at a lower level in DU-145 and at minimal levels in PC-3 cells as compared to PNT2 cells.

E-cadherin levels were consistent with their mRNA expression. (Note that E-cadherin appeared to be lost in PC-3 and DU-145 in this relative Western blotting analysis compared with completely epithelial PNT2 cells, although E-cadherin could be detected in PC-3 in different culture conditions as we have shown [9,13].) Vimentin was highly expressed in DU-145 cells and well-detectable in PC-3 and normal prostate cells, although not detectable in PNT2 cells.

ZO-1 (also known as tight junction protein 1: TJP1) was detected in epithelial cell line PNT2 and normal prostate cells at reduced levels in DU-145 and not detected in PC-3 cells. These data suggest that SCAND1 expression correlates with epithelial features, whereas the loss of SCAND1 is correlated with mesenchymal features.

Next, we examined whether SCAND1 and MZF1 expression levels were correlated in clinical prostate adenocarcinoma specimens. Indeed, SCAND1 and MZF1 expression was significantly correlated in prostate adenocarcinomas, indicating that these SCAN domain proteins are consistently co-expressed (Figure 1F).

These data suggest that SCAND1 expression is involved in maintaining epithelial features, whereas the simultaneous loss of SCAND1 correlated with mesenchymal and E/M phenotypes in tumor cells.

### 3.2. SCAND1 Is Mutually Inducible with MZF1(ZSCAN6) and Localized in Hetero-Chromatin

It has been shown that the SCAN domain is an oligomerization domain with SCAN-TFs and other proteins [21,24,25,26]. We have shown that SCAND1-MZF1(ZSCAN6) oligomerized complexes were bound to ‘an MZF1-binding site’ in chromatin in the *CDC37* gene promoter region in prostate cancer cells [28]. To examine whether SCAND1 and MZF1 were mutually inducible and co-localized and binding to chromatin, we next transfected pcDNA3.1-MZF1^Flag^, pCMV6-SCAND1^Flag^, and pLV-SCAND1^HA3^ into DU-145 cells, and established stable clones (Figure 2A). SCAND1 overexpression induced MZF1 mRNA, while MZF1 overexpression induced SCAND1 mRNA in DU-145 cells, indicating the mutual inducibility of these SCAN-TFs (Figure 2B–E).

Overexpressing SCAND1^HA3^ was co-localized with MZF1 in nuclei (Figure 2F). Moreover, overexpressing SCAND1^Flag^ was co-localized with heterochromatin protein 1γ (HP1γ) ^HA3^, indicating that SCAND1 was involved in hetero-chromatin, which is a repressive chromatin (Figure 2G). SCAND1^HA3^ overexpression induced a cobblestone-like epithelial morphology with intercellular adhesion formed with β-catenin (Figure 2H).

We next performed Western blotting to examine the expression and chromatin-binding of MZF1 and SCAND1 in stable clones. Overexpressing SCAND1 was bound to chromatin in the SCAND1 c1, c2 and c3 clones (Figure 2I, top panel; Figure 2J). Soluble SCAND1 was lower in the SCAND1 c1 clone compared to c2 and c3 clones. (The difference might be caused by a lower concentration of G418 used for establishing the c1 clone). 

Overexpressing MZF1 was detected in chromatin and soluble fractions, while native soluble MZF1 and MZF1 potentially undergoing PTMs in chromatin were most detectable in the MZF1 c2 clone (Figure 2I, third and fourth panels; Appendix A and Figure 2K). (We previously reviewed MZF1 PTMs [25]). 

We next analyzed mutual inducibility and coordinated chromatin-binding of SCAND1 and MZF1. Overexpressing and chromatin-binding SCAND1 induced MZF1 expression and chromatin binding, suggesting that SCAND1 induced MZF1 expression and complex formation on the chromatin (Figure 2I, third panel; Figure 2K, Appendix A). MZF1 overexpression induced SCAND1 expression in the MZF1 c2 clone (Figure 2I, second panel; Figure 2J). Simultaneously, HP1γ, a heterochromatin protein, was bound in chromatin in the SCAND1 c1, c2, and c3, and MZF1 c2 clones (Figure 2I, third-panel from the bottom; Figure 2L), suggesting that mutually induced SCAND1 and MZF1 could involve heterochromatin formation with HP1γ.

These data indicated that SCAND1 and MZF1 are mutually inducible and coordinately bound to hetero-chromatin, repressive chromatin, with HP1γ.

### 3.3. SCAND1 Inhibits Tumor Cell Proliferation and Reverses EMT

We examined whether SCAND1 and MZF1 could alter tumor cell proliferation and EMT. Hybrid E/M status was observed in both vector-transfected and MZF1 overexpression DU-145 cells (Figure 3A). Of note, the overexpression of SCAND1 changed the DU-145 cells from the hybrid E/M status into a cobblestone-like epithelial morphology. Both SCAND1 and MZF1 overexpression significantly inhibited tumor cell proliferation, while SCAND1 overexpression more significantly inhibited it (Figure 3B).

We next performed Western blotting for E-cadherin, vimentin, phosphorylated MAPK-ERK1/2, and NF-κB to confirm the role of SCAND1 and MZF1 in regulating EMT and cell proliferation markers. E-cadherin was increased by SCAND1 overexpression (Figure 3C,D), whereas vimentin was reduced by SCAND1 overexpression (Figure 3C,E), suggesting that SCAND1 reversed EMT in DU-145 cells and induced an epithelial status.

Phosphorylated MAPK-ERK-1/2 levels were reduced by SCAND1 overexpression (Figure 3C,F). In addition, phosphorylated NF-κB p65 (also known as RelA) was reduced by SCAND1 overexpression (Figure 3C,G). HSP90β, a constitutively expressed type of HSP90 often used as a loading control, was not altered by overexpression studies.

We next examined whether MZF1 regulated *VIM* gene expression, a mesenchymal marker. The overexpression of MZF1^Flag^ upregulated *VIM* mRNA expression (Figure 3H), while siRNA targeting MZF1 downregulated it (Figure 3I), suggesting that MZF1 positively regulates *VIM* gene expression.

To verify whether the subcellular localization of β-catenin and E-cadherin were altered by SCAND1 overexpression along with epithelialization, we used confocal microscopy for immunocytochemistry. β-Catenin was accumulated in the nuclei of the non-transfected and MZF1-overexpressed cells, while this versatile protein was also localized at intercellular adhesin sites in the hybrid E/M status (Figure 3J). In contrast, SCAND1 overexpression appeared to reduce nuclear β-catenin and led to relocation to intercellular adhesion sites formed with E-cadherin (Figure 3J,K). We also confirmed that SCAND1^HA3^ induced E-cadherin intercellular adhesion by lentivirus-based overexpression (Figure 3L). In addition, nuclear E-cadherin was found in SCAND1 overexpressing cells (Figure 3J–L).

These data suggested that SCAND1 overexpression induced functional E-cadherin that formed epithelial intercellular adhesion, transduced prostate tumor cells into epithelial status, and reduced tumor cell proliferation by inhibiting the MEK-ERK signaling pathway.

### 3.4. SCAND1 and MZF1 Expression Negatively Correlates with Gene Expression of MAP3Ks (MEKKs), MAPKs, and EMT Drivers 

We reported that MZF1 and SCAND1 formed oligomers in chromatin and were bound with an MZF1-binding site in the promoter region of *CDC37* gene, indicating MZF1/SCAND1 hetero-oligomers as transcriptional repressor complex [28]. We also have shown that SCAND1 and MZF1 could be co-expressed and co-localized in nuclei in prostate cancer cells in Figure 2. To determine whether SCAND1-MZF1 could be involved in regulating the gene expression of MAPKs and EMT drivers, we next investigated co-expression signatures of SCAND1 and MZF1 versus gene expression of MAPKs (including MAP3Ks) and EMT drivers, including CTNNB1 (encoding β-catenin), ZEB1 and ZEB2, and TGFBRs (encoding TGFβ receptors). 

Of note, we found that SCAND1 and MZF1 expression negatively correlated with expression of two MAP3K genes (MAP3K1(MEKK2) and MAP3K2(MEKK1)) and two MAPK genes (MAPK1(ERK2) and MAPK14(p38Alpha)) (Figure 4A,B; Appendix A). These data suggest that SCAND1-MZF1 could mediate versatile repression of the MAP3Ks (MEKK1 and MEKK2) genes, which are essential for the phosphorylation of MAPKKs (MEK) and subsequent ERK.

Regarding EMT drivers, both SCAND1 and MZF1 expression negatively correlated with CTNNB1 expression in prostate tumor specimens (Figure 4C). Moreover, SCAND1 and MZF1 expression negatively correlated with ZEB1 and ZEB2 expression (Figure 4D). These data suggest that SCAND1-MZF1 could repress ZEB1, ZEB2, and CTNNB1 genes, an effect that might reverse the EMT phenotype, consistent with the data from cell biology in Figure 2. Furthermore, SCAND1 and MZF1 expression was negatively correlated with TGFBR1, TGFBR2, and TGFBR3 gene expression (Figure 4E), suggesting that SCAND1-MZF1 could reduce TGFβ receptors to narrow down TGFβ signals emanating from the microenvironment to tumor cells.

### 3.5. SCAND1 Overexpression Inhibited Collective Migration and Lymph-Node Metastasis of Prostate Cancer

Tumor cell migration and invasion are crucial processes in the initiation of metastasis, and hybrid EMT is known to be involved in these processes [18,49]. Metastatic prostate cancer often disseminates to pelvic lymph nodes [13,50]. Recent studies have shown that tumor budding of 5–20 cells collectively migrated and invaded stroma, a process known as collective migration [51,52].

We have recently shown that SCAND1 overexpression could inhibit the tumor growth of prostate cancer cells in vivo [28]. We next used a mouse tumor xenograft model to examine whether SCAND1 could inhibit tumor cell EMT, migration, and metastasis. Enhanced co-expression of E-cadherin was seen in/around the cells overexpressing SCAND1, indicating that SCAND1 influenced the tumor cells towards an epithelial phenotype (Figure 5A). Vimentin was abundantly expressed in the control tumors, although significantly reduced in the SCAND1-overexpressing cells (Figure 5B–D). The vimentin-positive cells were spindle-shaped in morphology, whereas the vimentin-negative cells had a cobblestone-like morphology (Figure 5C). These data suggest that SCAND1 reversed EMT in prostate tumor cells to a more epithelial status in vivo.

Ki-67 expression, a marker of proliferating tumor cells, was significantly suppressed by SCAND1 overexpression in in vivo tumors (Figure 5E,F). Of note, the collective migration of Ki-67-positive cancer cells at an invasive front was seen budding from the tumor to invade stroma in the control tumors but not in SCAND1-overexpressing tumors (Figure 5E, bottom left panel, arrowheads).

Next, we examined whether SCAND1 overexpression in prostate tumors could alter pelvic lymph node metastasis in the mouse xenograft model. Lymph node metastases were found in 100% (4/4) of the control mice, although reduced to 25% (1/4) in the SCAND1-overexpressing tumor xenograft mice (Figure 5G). The average rate of pelvic lymph node metastasis was 70% in the control mice versus 25% in the SCAND1-overexpressing tumor xenograft mice. 

These data suggest that SCAND1 overexpression inhibited EMT, collective migration, and lymph-node metastasis of prostate cancer in vivo.

### 3.6. SCAND1 and MZF1 Expression Correlated with the Prognosis of Patients Suffering from Cancers

To clarify the prognostic values of MZF1 and SCAND1, we next investigated whether MZF1 and SCAND1 high or low expression correlated with survival rates of patients suffering from pancreatic, head and neck, kidney, and prostate cancers. High expression of MZF1 and SCAND1 correlated with better prognosis in pancreas DAC and head and neck SCC (Figure 6A, upper 4 panels; Table 2). On the other hand, high expression of MZF1 and SCAND1 correlated with poorer prognosis in RCC (Figure 6A, third row; Table 2). Moreover, high expression of MZF1 correlated with poorer prognosis in prostate adenocarcinoma (Figure 6A, bottom). Meanwhile, Kaplan–Meier graphs of SCAND1 high versus low expression cross each other, and there was no correlation with the prognosis of prostate adenocarcinoma.

These data suggest that SCAND1-MZF1 high expression is a better prognosis marker for patients suffering from pancreatic cancer and head and neck cancers, whereas SCAND1-MZF1 high expression is a poorer prognosis marker for patients suffering from kidney cancer. MZF1 high expression is a poorer prognosis marker for patients suffering from prostate adenocarcinoma.

## 4. Discussion

In the present study, we showed for the first time that SCAND1 reverses EMT, inhibits tumor cell proliferation, and reduces the invasive capacities of prostate cancer cells. The anti-EMT and tumor-suppressive effects of SCAND1 overexpression in DU-145 cells are consistent with our previous report showing that SCAND1 overexpression inhibited tumor growth of PC-3 prostate cancer cells in vivo [28]. We showed that SCAND1 and MZF1 are mutually inducible, localized in heterochromatin with HP1γ (Figure 2), spatially co-localized in cancer cells in vitro, and coordinately expressed in many cases of clinical prostate adenocarcinomas (Figure 1). These findings are consistent with previous reports showing that SCAND1 and SCAN-ZFs, including MZF1, form hetero-oligomers co-repressing transcription [24,26,28]. Overexpression of SCAND1 induced expression and co-localization of MZF1 and reversed partial EMT status into an epithelial status, while SCAND1 and MZF1 expression negatively correlated with gene expression of EMT drivers, including ZEB1, ZEB2, TGFβ receptors and β-catenin (Figure 2, Figure 3, Figure 4 and Figure 5). These data suggested that SCAND1-MZF1 complexes can inhibit EMT by repressing *ZEB1/2*, *TGFBRs*, and *CTNNB1* (Figure 7). Moreover, the overexpression of SCAND1 inhibited the phosphorylation of ERK-1/2 and tumor cell proliferation (Figure 3). Consistently, SCAND1 and MZF1 expression negatively correlated with gene expression of MAP3Ks (MEKK-1/2) and ERK-MAPK (Figure 4). These data suggest that SCAND1-MZF1 complexes can suppress tumor cell proliferation by repressing MEKK/MAP3K genes and quenching the MEK-ERK signaling pathway (Figure 7).

Our data also touch upon the prognostic importance of SCAND1 and MZF1 expression in evaluating several types of cancer. MZF1 and SCAND1 high expression correlated with better prognoses in patients suffering from pancreatic cancer and head and neck cancers (Figure 6), suggesting that elevated SCAND1-MZF1 co-expression could inhibit EMT and invasive phenotypes in cancer. It was consistently reported that the downregulation of MZF1 is associated with gastric tumourigenesis, suggesting that MZF1 could be an early predictive and prognostic biomarker of better prognosis in cancer patients [53]. On the other hand, SCAND1 and MZF1 high expression correlated with poor prognosis in patients suffering from kidney cancer. Moreover, MZF1 high expression correlated with poor prognosis in patients suffering from prostate cancer. Consistently, we have shown that the MZF1–CDC37 axis is intensely active in prostate cancer and involves cancer progression [28]. Nevertheless, the current analyses could not find any prognostic value of SCAND1 expression in clinical prostate cancer.

A therapeutic strategy to transduce mesenchymal tumor cells into epithelial cells by inducing SCAND1-MZF1 could potentially help cancer therapy. Epithelial tumor cells are more sensitive to drugs, responsive to apoptosis signals, and susceptible to immune attack than mesenchymal tumor cells, whereas mesenchymal tumor cells are more chemoresistant [19]. However, if not completed, a tumor epithelialization strategy might be context-dependent, as partial EMT might induce tumor cell stemness, tumor initiation capacity, and adaptation to changes in the microenvironment and metabolism, which could generate resistant cancer recurrence that is difficult to treat [18]. Indeed, tumor cell spheroids are formed with epithelial adhesion by E-cadherin and EpCAM while simultaneously expressing mesenchymal marker vimentin, indicating that a hybrid/partial EMT status may be essential for tumor and spheroid formation [13]. Moreover, spheroid formation involves enhanced stemness, chemoresistance, and malignant exosome release in cancer [13,54,55,56,57]. Our data indicate that elevated expression of SCAND1 and MZF1 can reverse the hybrid EMT status of tumor cells to a more epithelial status, although vimentin and nuclear β-catenin remained, suggesting that the cells were not fully epithelial. These could be the limitations of our current study. However, our data indicate that SCAND1 overexpression inhibited migration, lymph node metastasis (Figure 5) and tumor growth in prostate cancer cells in vivo [28].

Our study also suggests that nuclear E-cadherin might be involved in reverse EMT. We showed nuclear E-cadherin in the SCAND1-overexpressing cells with reverse EMT status (Figure 3). It was reported that the cleaved cytoplasmic domain of E-cadherin was translocated in the nucleus in epithelial cell lines (A431, MCF-7, and MDCK) [58] and in a metastatic colorectal cancer model [59]. Then, it was reported that an aberrant nuclear localization of E-cadherin is a potent inhibitor of Wnt/β-catenin-elicited promotion of the cancer stem cell phenotype [60]. On the other hand, it was reported that nuclear E-cadherin acetylation promoted colorectal tumorigenesis via enhancing β-catenin activity. In our current study, nuclear E-cadherin was seen in the plasmid-based SCAND1 overexpressing cells, although less found in lentivirus-mediated SCAND1 overexpressing cells. Nevertheless, it will be important to determine how nuclear E-cadherin and SCAND1 regulate EMT and cancer stem cells.

Our study also suggests that the SCAND1-MZF1 complex is pleiotropic and can counteract tumor microenvironmental factors released by CAFs and control several signaling pathways that activate EMT (Figure 4). These include the TGFβ signal, Wnt/β-catenin, and the MAPK signaling pathway. TGFβ plays a central role in inducing EMT in several different tissue and tumor types [61,62,63]. The TGFβ-driven EMT activation mechanism collaborates with several other signaling systems, including the MAPK (MEKK-MEK-ERK/MAPK and p38-MAPK) and PI3K-AKT-NF-κB pathways that also contribute to activating the EMT programs [61]. The MEK-ERK signaling pathway mediates the tumor microenvironmental EMT-inducing mitogenic signals, such as EGF, FGF, extracellular HSP90, and extracellular vesicles (EVs) [10,64,65]. Therefore, the SCAND1-based gene repression of MAP3Ks and MAPKs can reduce mitogenic and EMT-inducing signals from EGFR and FGFR. Our data also suggest that SCAND1 could suppress the WNT/β-catenin signal, which is crucial for EMT and cancer stemness. Therefore, SCAND1-based gene repression of CTNNB1 can minimize EMT-TF gene expression and thus reverse EMT. 

Our study also touches upon a mechanism by which SCAND1 represses gene expression involving EMT. Our data indicate that SCAND1 and MZF1 were co-localized to heterochromatin with HP1, suggesting that SCAND1-MZF1 complexes involve forming a repressive chromatin status (Figure 2). Consistently, we have shown that overexpressing SCAND1 and MZF1 form oligomers in chromatin and bind to an MZF1-binding site in the promoter region of *CDC37* gene, and hetero-oligomers of SCAND1 and MZF1 can repress *CDC37* gene expression in prostate cancer [28]. (A graphical abstract showing hetero-oligomerization of MZF1 and SCAND1 was published in a supplemental figure.) The oligomerizing function of the SCAN domain was first reported by Tucker Collins’ group in the paper entitled "the zinc finger-associated SCAN box is a conserved oligomerization domain" [24]. Subsequently, Tara Sander et al. identified a novel SCAN box-related protein called RAZ1 (also known as SCAND1) that interacts with MZF1B to show the leucine-rich SCAN box mediated hetero- and homo-protein associations [26]. Thus, SCAN-ZFs, including MZF1, can mediate the repressive role of SCAND1 in gene regulation.

Our data also suggest that HP1-based heterochromatin formation is induced by SCAND1 overexpression (Figure 2). The HP1 family is composed of HP1α/CBX5, HP1β/CBX1, and HP1γ/CBX3, which could be key cofactors of SCAND1 to repress gene transcription. However, HP1γ expression is elevated in prostate cancer and is superior to the Gleason score as a predictor of biochemical recurrence after radical prostatectomy [66]. Moreover, HP1γ is upregulated in tongue SCC, breast cancer, and colorectal cancer (CRC) and is associated with an unfavorable prognosis [67,68,69]. The overexpression of HP1γ in pancreatic DAC promotes cell cycle transition-associated tumor progression [70]. HP1γ expression is higher in tumor tissues than paired normal tissues in kidney cancer [71], CRC [69], gastric cancer [72], and non-small cell lung cancer [73]. These unfavorable roles of HP1γ might be caused by its transcriptional activator function [74,75] collaborating with nuclear moonlighting metalloproteinases (MMP) [39,40,76] or RNA processing [77]. Tumor microenvironmental MMPs mediate intercellular communications via EVs that deliver MMPs into cellular nuclei to bind with HP1, leading to the gene activation of cellular communication network factor 2 (CCN2) [43,78,79]. Moreover, HP1γ plays a critical role in reprogramming to pluripotency [80], which is involved in cancer stemness and the activation, inhibition and hybrid of EMT. Our data indicate that HP1γ is involved in the chromatin localization and function of SCAND1 and MZF1 (Figure 2), while HP1α and HP1β might also collaborate with SCAND1 and MZF1. Our overexpression study suggests that the co-expression of SCAND1 and SCAN-ZFs can contribute to heterochromatin formation for the repression of oncogenes. However, the target genes may also include tumor suppressor genes.

Our study also suggests that PTMs and hotspot mutations are involved in the functions of MZF1 and SCAND1. The overexpression of SCAND1 and MZF1 increased 15–35kD upshifted bands of MZF1 (Figure 2). We experienced similar upshifts previously as well [28]. MZF1 contains five sumoylation sites (Appendix A). Although the theoretical molecular weight of the SUMO proteins is approximately 11 kDa, the size increase for each SUMO added on SDS-PAGE is typically in the range of 15–17 kDa. Therefore, MZF1 might undergo mono-sumoylation and di-sumoylation in our study (Figure 2). MZF1 also contains two SIM (Appendix A) [25]. Therefore, MZF1 could form oligomers through SUMO-SIM interactions and inter-SCAN domain interactions. Some sumoylation might be induced by MZF1 phosphorylation. MZF1 and SCAND1 contain ubiquitination sites in their SCAN domains (Appendix A). Ubiquitination of SCAN domains could promote their degradation and thus inhibit their oligomerization. These PTMs and their modifier enzymes can regulate the function of MZF1 and SCAND1. Moreover, 3D-clustered hotspot mutations were found in the SCAN domain of MZF1 in patient-derived tumor specimens (Appendix A). These mutations can alter the structure and functions of MZF1 in cancer.

In addition to their target gene *CDC37*, we also have unpublished data demonstrating MZF1 and SCAND1 localized to an MZF1-binding site in the promoter region of *HSP90* gene, which encodes a partner of CDC37. We showed the molecular chaperones CDC37 and heat shock protein 90 (HSP90) and HSP90-rich exosomes to be crucial in promoting EMT in prostate cancer and tongue cancer and initiating EMT in normal epithelial cells [9,10,12]. Therefore, it is proposed that the mechanism by which SCAND1-MZF1 complexes reverse EMT might involve the transcriptional co-repression of *CDC37* and *HSP90* genes and a reduction in exosome secretion. The exact mechanisms should be clarified in the future. Our data suggest that SCAND1 and MZF1 might also regulate molecular chaperone expression and the cell stress response in cancer. Cell stresses, such as heat shock stress, activate heat shock transcription factor 1 (HSF1) that induces stress-responsive chaperone expression, including HSP90 and CDC37 [9]. Subsequently, the molecular chaperone CDC37 promotes exosome secretion [9]. Thus, it is suggested that transcription factors HSF1, MZF1, and SCAND1 co-regulate cell stress response, molecular chaperone gene expression, and exosome secretion. SCAND1 and MZF1 could also be involved in the cell stress response by crosstalking with HSF1. Detailed mechanisms should be clarified in the future.

In conclusion, our current study suggests that SCAND1 induces expression and coordinated chromatin-binding of MZF1 to reverse the hybrid E/M status into epithelial phenotype and inhibits tumor cell proliferation, migration, and metastasis, potentially by repressing the gene expression of EMT drivers and the MAP3K-MEK-MAPK signaling pathway. 

## Figures and Tables

**Figure 1 cells-11-03993-f001:**
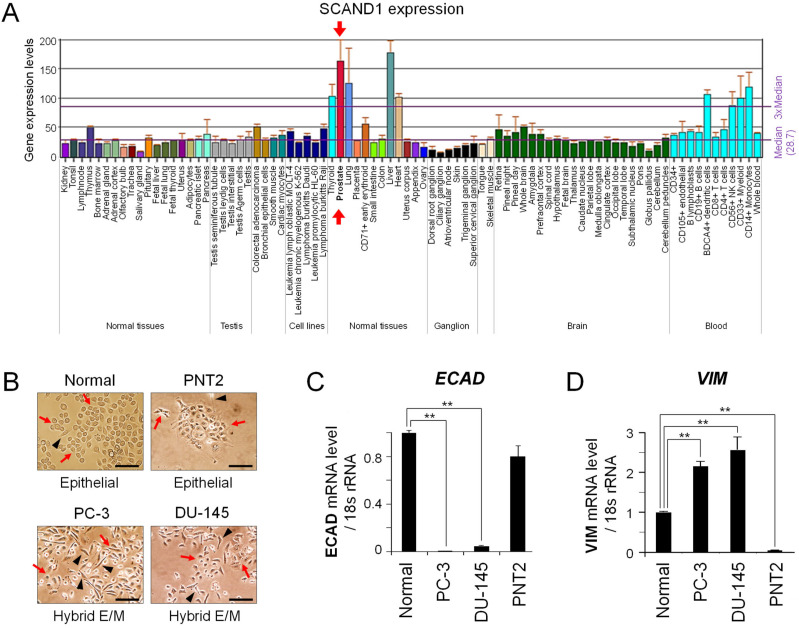
SCAND1 expression is involved in epithelial features of the prostate. (**A**) SCAND1 gene expression in various tissues. Data were obtained from BioGPS database. (**B**) Morphologies of normal prostate cells, PNT2, PC-3, and DU-145 cells. Red arrows indicate polygonal, cobblestone-like epithelial cells. Arrowheads indicate spindle-shaped mesenchymal cells. Scale bars, 100 μm. (**C**,**D**) *ECAD* and *VIM* mRNA expression evaluated by qRT-PCR. ** *p* < 0.01 (versus Normal), *n* = 3. (**E**) Western blotting of SCAND1, MZF1, E-cadherin, ZO-1, and vimentin. Actin, loading control. Blots of SCAND1 and actin were also published in ref [28]. (**F**) Co-expression correlation analysis of *SCAND1* and *MZF1* mRNA in prostate adenocarcinoma specimens.

**Figure 2 cells-11-03993-f002:**
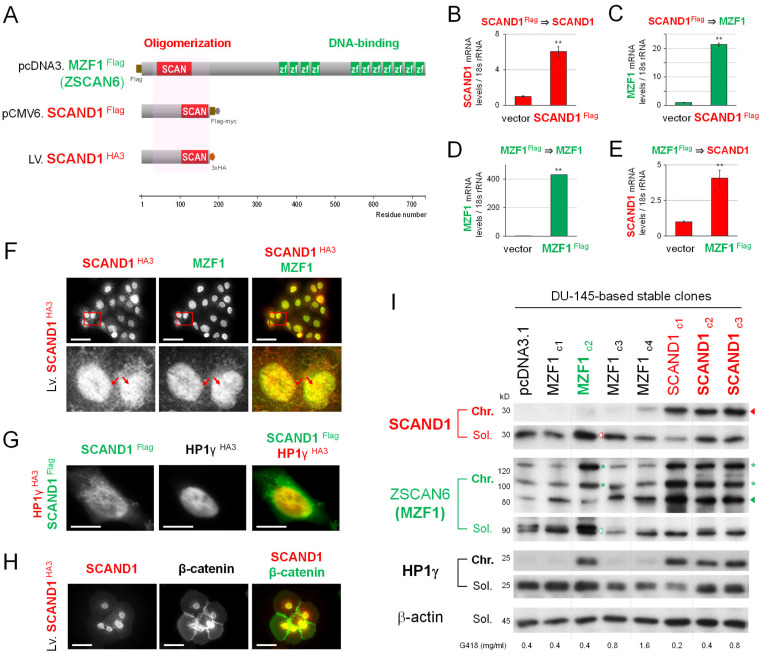
SCAND1 is mutually inducible with MZF1(ZSCAN6) and localized in hetero-chromatin. (**A**) Constructs for overexpressing MZF1^Flag^, SCAND1^Flag^, and SCAND1^HA3^. These were transfected into DU-145 cells, and stable clones were established. (**B**–**E**) qRT-PCR analysis of SCAND1 and MZF1 in the stable clones. ** *p* < 0.01, *n* = 3. (**F**–**H**) Confocal microscopy showing co-expression and co-localization between; (**F**) SCAND1^Flag^ vs. MZF1. Antibodies against HA-tag and MZF1 were used. Scale bars, 50 μm. Arrows indicate speckles of SCAND1/MZF1. (**G**) SCAND1^Flag^ vs. HP1γ^HA3^. Overexpression plasmids were co-transfected into HeLa cells. Antibodies against Flag-tag and HA-tag were used. Scale bars, 10 μm. (**H**) SCAND1 vs. β-catenin. Antibodies against SCAND1 and β-catenin were used. Scale bars, 50 μm. (**I**) Western blotting showing SCAND1, MZF1(ZSCAN6), and HP1γ in chromatin (Chr.) and soluble (Sol.) fractions in stable clones. Arrowheads indicate native full-length proteins. Asterisks indicate MZF1 undergoing PTMs. (**J**–**L**) Expression levels of SCAND1, MZF1, and HP1γ in stable clones. Relative intensities of bands in western blotting were quantified. * *p* < 0.05, *n* = 3. ** *p* < 0.01, *n* = 3. *** *p* < 0.005, *n* = 3.

**Figure 3 cells-11-03993-f003:**
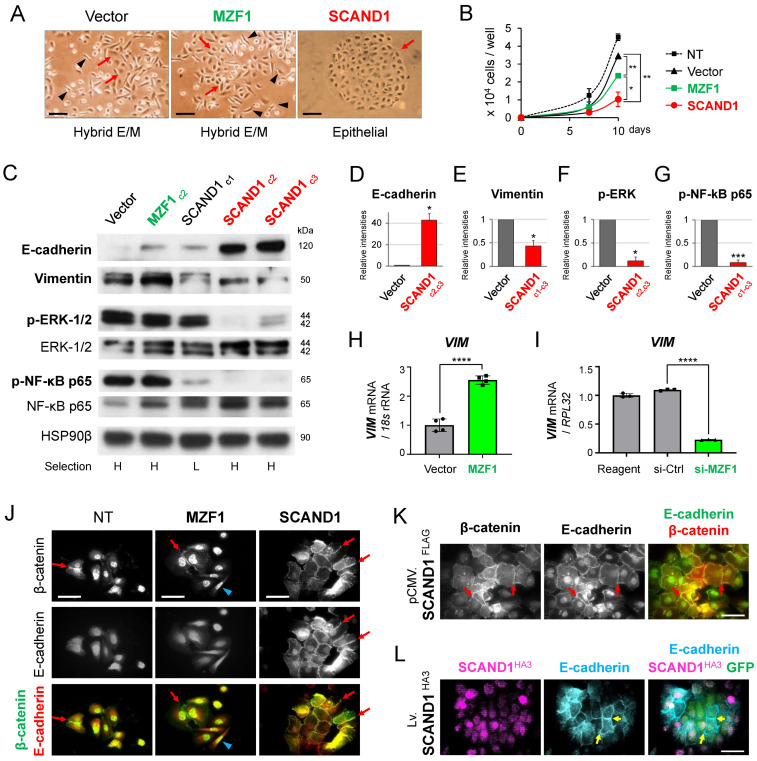
SCAND1 inhibits tumor cell proliferation and reverses EMT to establish epithelial adhesion. Plasmids for overexpressing SCAND1 ^Flag^ and MZF1 ^Flag^ or the control pcDNA3.1 vector were transfected into DU-145, and stable cells were cloned. (**A**) Morphologies of SCAND1- or MZF1-overexpressing and control vector-transfected DU-145 cells. Arrowheads indicate spindle-shaped mesenchymal cells. Red arrows indicate polygonal, cobblestone-like epithelial cells. Scale bars, 100 μm. (**B**) Cell proliferation curves. NT, non-transfected cells. * *p* < 0.05, ** *p* < 0.01. (**C**) Western blotting showing E-cadherin, Vimentin, phosphorylated or total ERK-1/2 and NF-κB p65. See Figure 2 for MZF1 and SCAND1 overexpression. HSP90β, loading control. (**D**–**G**) Relative expression levels. Band intensities of Western blotting were quantified. *** *p* < 0.005, *n* = 3. * *p* < 0.05, *n* = 2 or 3. (**H**,**I**) *VIM* mRNA expression altered by MZF1 expression plasmid (**H**) or siRNA (**I**) transfected in DU-145 cells. **** *p* < 0.001, *n* = 4. (**J**–**L**) Confocal microscopy showing subcellular co-localization of β-catenin vs. E-cadherin (**J**,**K**) and SCAND1 ^HA3^ vs. E-cadherin (**L**). Arrows indicate intercellular adhesion. Blue arrowheads indicate spindle-shaped cells. Scale bars, 100 μm.

**Figure 4 cells-11-03993-f004:**
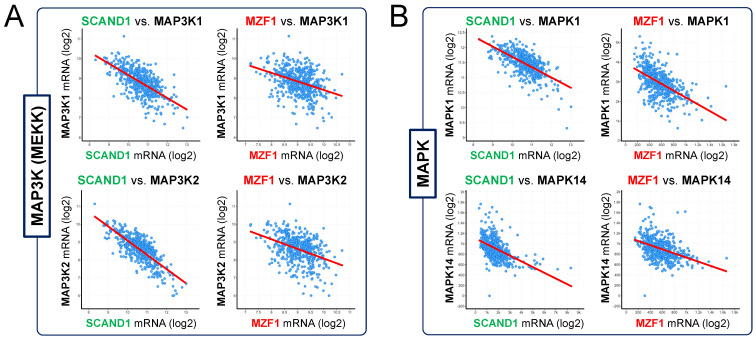
SCAND1 and MZF1 expression negatively correlated with gene expression of MAP3Ks (MEKKs), MAPKs, and EMT drivers in prostate tumors. We analyzed the co-expression of SCAND1 and MZF1 vs. (**A**) MAP3K1 (MEKK1) and MAP3K2 (MEKK2), (**B**) MAPK1 (ERK2) and MAPK14 (p38Alpha), (**C**) CTNNB1 (encoding β-catenin), (**D**) ZEB1 and ZEB2, and (**E**) TGFβ receptors (TGFBR1, TGFBR2, and TGFBR3) in patients-derived prostate adenocarcinoma specimens.

**Figure 5 cells-11-03993-f005:**
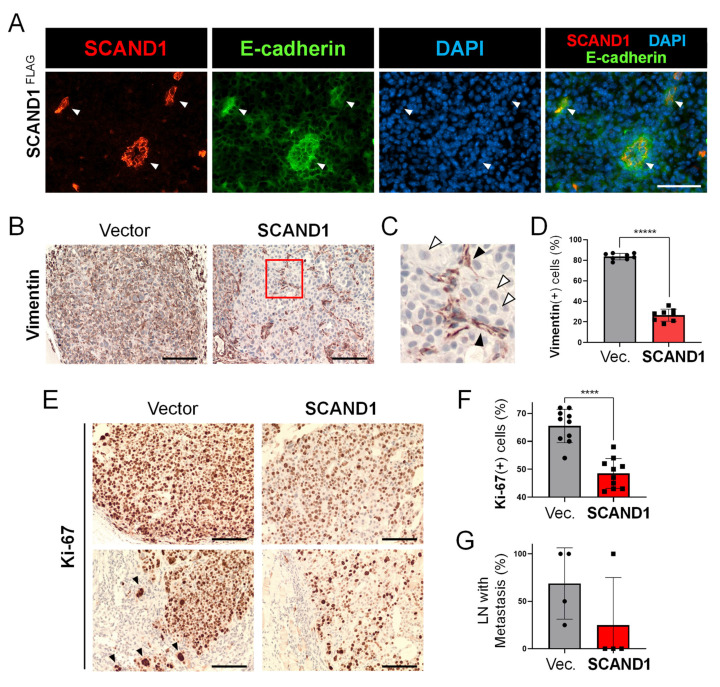
SCAND1 inhibits EMT, collective migration, and lymph-node metastasis in vivo. SCAND1-overexpressed or the control vector-transfected DU-145 cells were subcutaneously transplanted into mice. Tumors were resected, and IHC was performed. (**A**) Co-expression of SCAND1 and E-cadherin in the tumor overexpressed with SCAND1. Arrowheads indicate co-expression. Scale bars, 50 μm. (**B**,**C**) IHC for Vimentin. (**C**) A magnified image showing vimentin(+) cells (black arrowheads) and Vimentin(-) cells (white arrowheads) in the SCAND1 overexpressing tumor. Scale bars, 50 μm. (**D**) Column scatter plotting showing the rate of vimentin(+) cells. ***** *n* = 8 fields, *p* < 0.000000000001. (**E**) IHC for Ki-67 in the center of tumors (upper panels) and tumor–stroma border areas (lower panels). Arrowheads indicate collectively migrating cancer cells at an invasive front into the stroma. Scale bars, 50 μm. (**F**) The rate of Ki-67(+) cells. **** *n* = 10 fields, *p* < 0.0001. (**G**) The rate of lymph nodes with metastasis in each mouse. *n* = 4 mice.

**Figure 6 cells-11-03993-f006:**
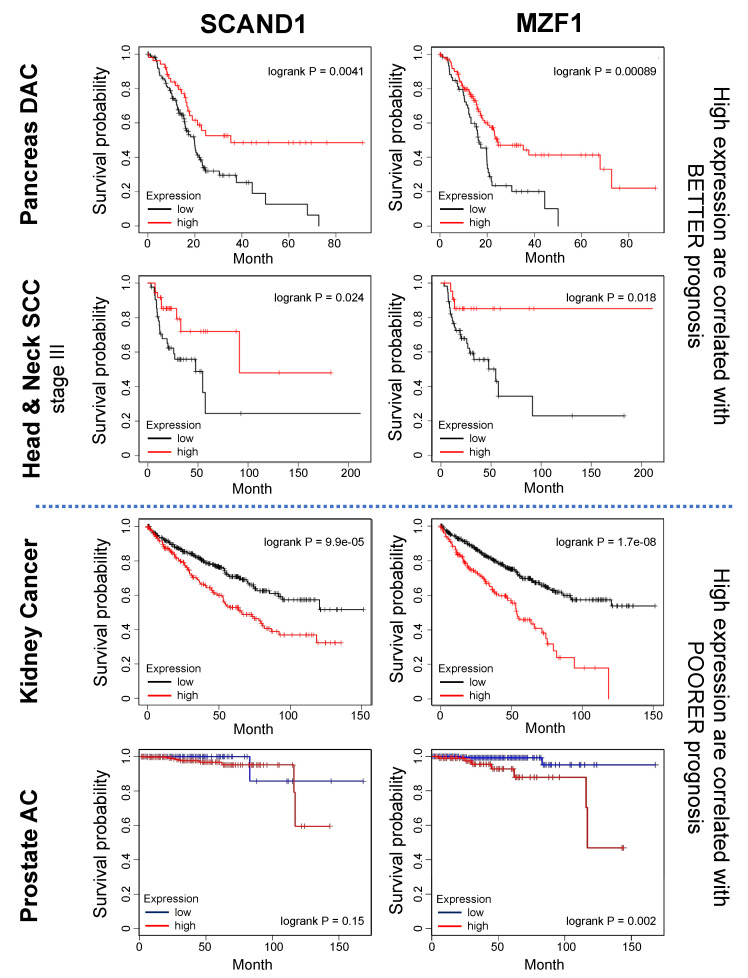
Prognostic values of SCAND1 and MZF1 expression in cancer patients. Kaplan–Meier plotting was performed to compare the prognosis of high- vs. low-expression groups of SCAND1 and MZF1 in patients suffering from pancreas cancer, head and neck cancer, kidney cancer (RCC), and prostate cancer.

**Figure 7 cells-11-03993-f007:**
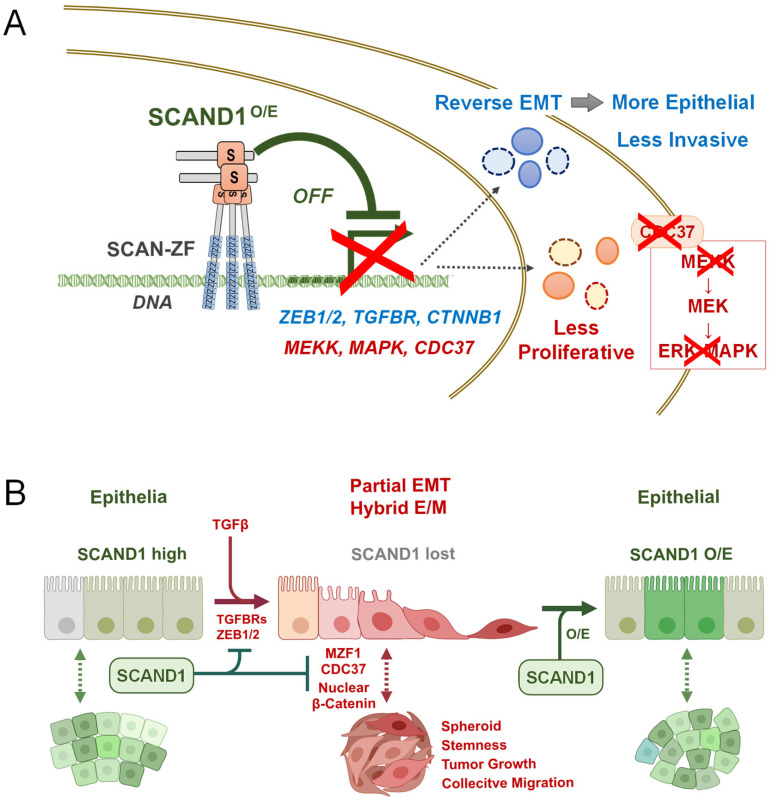
SCAND1 reverses hybrid E/M tumor cells to a more epithelial, less invasive status and inhibits their proliferation, migration, and metastasis. (**A**) Transcriptional repression by SCAND1 overexpression (O/E). The combination of SCAND1 and SCAN-ZF, such as MZF1, can repress EMT driver genes, including ZEB1/2, TGFBRs, and CTNNB1, to reverse EMT for establishing a more epithelial and less invasive phenotype. SCAND1 can also repress genes encoding kinases in MEKK-MEK-ERK/MAPK signaling pathway to inhibit tumor cell proliferation. (**B**) Partial EMT (or hybrid E/M) is controlled by SCAND1. SCAND1 maintains the epithelial status of cells, whereas the loss of SCAND1 may initiate EMT. SCAND1 O/E can reverse EMT to retrieve epithelial status.

**Table 1 cells-11-03993-t001:** Primer sequences for qRT-PCR.

Primer Name	Sequences (5′ to 3′)
h ECAD 1937F	AGG AAT CCA AAG CCT CAG GT
h ECAD 2065R	TTG GGT TGG GTC GTT GTA CT
h VIM 892F	AGG TGG ACC AGC TAA CCA AC
h VIM 1010R,	GGC TTC CTC TCT CTG AAG CA
h SCAND1 F#1	CGC AGA GAA GCC AGA GAC TT
h SCAND1 R#10	TCA GCA CTG CGT CTG CAC C
h MZF1-F785	TGC AGG TGA AAG AGG AGT CA
h MZF1-R939	AGT CTT GCT GTG GGG AAA GA
RPL32 F	CAG GGT TCG TAG AAG ATT CAA GGG
RPL32 R	CTT GGA AAC ATT GTG AGC GAT C
18s rRNA-h1245F	GAC TCA ACA CGG GAA ACC TC
18s rRNA-h1364R	AGA CAA ATC GCT CCA CCA AC

**Table 2 cells-11-03993-t002:** Prognostic values of SCAND1 and MZF1 expression in cancer.

Cancer Type	Log-Rank P		Hazard Ratio		Correlation with SCAND1 and MZF1 High Expression
	SCAND1	MZF1	SCAND1	MZF1	
Pancreas DAC	0.0041 **	0.0009 **	0.49	0.5	Better prognosis
Head & Neck SCC, Stage III	0.024 *	0.018 *	0.40	0.26	Better prognosis
Renal Cell Carcinoma	9.9 × 10^−5^ ***	1.7 × 10^−8^ ***	1.79	2.38	Poorer prognosis
Prostate Adenocarcinoma	0.15	0.002 **	0.21	0.0062	MZF1: poorer prognosis

* *p* < 0.05, ** *p* < 0.05, *** *p* < 0.0001.

## Data Availability

TCGA PanCancer Atlas and co-expression data are available in cBioPortal (cbioportal.org). Kaplan-Meier data sets are available in KM plotter (kmplot.com) and GEPIA2 (gepia2.cancer-pku.cn). Tissue-specific gene expression data are available in BioGPS portal (biogps.org). PTM data are available in PhosphoSitePlus (phosphosite.org).

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
