# Peer review of "SCAND1 Reverses Epithelial-to-Mesenchymal Transition (EMT) and Suppresses Prostate Cancer Growth and Migration"

_cells, 2022, doi:10.3390/cells11243993_

Round 1

Reviewer 1 Report

The manuscript reports on the role of SCAND1 in EMT and tumor growth and aggressiveness. The authors use a prostate cancer cell line model including induction of tumor xenografts for functional analysis. They also analysed published expression data sets for co-expression and survival.

The data and presentation are mostly clear and convincing and I have only minor complaints.

Specifically:

·        - Figure 4 shows the IHC-analysis of tumor xenografts, but misses the quantification of the staining intensity / density. This should be included.

·        - Figures 3 and 5 / table 1: can the authors supply references for the source of the analysed data sets beyond the Data Availability Statement given at the end of the manuscript?

·        - Figure 3 and table 1 are largely redundant. Therefore, one should be removed. The same is the case for Figure 5 and table 2.

Author Response

Thank you for your important comments. Accordingly, we revised the manuscript as follows.

-Figure 4 shows the IHC-analysis of tumor xenografts, but misses the quantification of the staining intensity / density. This should be included.

Accordingly we included quantification and statistics of Vimentin and Ki-67 staining (current Fig 5).

- Figures 3 and 5 / table 1: can the authors supply references for the source of the analysed data sets beyond the Data Availability Statement given at the end of the manuscript?

Accordingly, we supplied references for the source of the analyzed data sets in the method section: A data set of prostate adenocarcinomas (Project ID: TCGA-PRAD, dbGaP study accession: phs000178, PanCancer Atlas; 494 patients/samples) were analyzed with Spearman’s rank correlation coefficient of co-expression using cBioPortal {Vargas, 2022;Gao, 2013}.     

- Figure 3 and table 1 are largely redundant. Therefore, one should be removed. The same is the case for Figure 5 and table 2.

Accordingly, we moved table 1 to supple. As we believe table 2 is small and important to emphasize the differences in P values and HR between cancer types, we remained it.

We appreciate this reviewer for important points.

Reviewer 2 Report

Eguchi etal in this manuscript investigated the roles of SCAND1 in cancer cells. However, they demonstrated them by only overexpression experiments. At the present time, loss of function analysis is absolutely required. In addition, the authors’ statement in the main text are frequently incorrect. Thus, this manuscript is suboptimal for publication in biological journals. Specific comments that should be addressed are listed.

1. Grammatical errors; 

  line 225, A data set of prostate adenocarcinoma(...) were << was? 

  line 229, (CTNNB1) were << was? 

  Line 319, E-cadherin were << was? And more

2.The authors’ statements are not correct.

  1) Line162, the pellet was stored.... The sentence should be deleted.

  2) Line261-262, expression levels of MZF1 of Normal cell (left side) is almost similar to those of DU145 cells (2nd from right side) in figure 1D.

  3) line266-267, among 4 cells used in figure 1D, the authors’ statements are supported by only Pc3 and PNT2 cells, but not other 2 cells.  

  4) line 347, figure 1 clearly indicates “colocalization between SCAND1 and MZF1, but not “direct interaction”. If the authors show it, experiments using recombinant proteins are needed. 

3. Cell morphology of normal cells (Figs. 1B, 1C) should be also shown in figure 1A.

4. Figure 1F clearly indicated DU-145 cells show 100% infection efficiency. Is it true? Show specificity of both antibodies using siRNAs etc. It remains unclear why anti-SCAND1 antibody is not used in these experiments. In addition, detailed information of anti-MZF1 antibody is missing. 

5. What is significance of figure 2B? Rather than figure 2B, expression of both SCAND1 and MZF1 should be shown in panels of figure 2E.

6. The authors failed to show the cell morphology of MZF1-overexpressed cells. Show it in figure 2C.

7. E-cadherin in DU-145 cells is almost nothing (Fig 1D), but detected in fig. 2F. In addition, the authors mentioned “E-cadherin faintly detectable in the cytoplasm and possibly in nuclei (line 324-326). Why is it not detected in WB analysis in figure 1D. Since non-specific staining in immunohistochemical study are frequently observed, the authors should exclude this possibility.

8. What do the authors want to claim on SCAND1 c1 cells (Fig 2E)? Phenomena in SCAND1 c1 are not completely different from those in other overexpressed cells. Moreover, as mentioned above, expression levels of SCAND1 should be shown.

9. Line 295-296; “the intracellular effects of SCAND1, which lacks a DNA-binding domain, are mediated by binding a SCAN-ZF partner, most likely MZF1 and forming a repressive heterodimer.” 

In general, transcription factor that lacks DNA binding domain acts as a dominant negative factor (Curr Top Dev Biol. 2014;110:189-216. doi: 10.1016/B978-0-12-405943-6.00005-1). Thus, it seems that MZF1 forms homodimer to directly bind to the DNA and acts a transcriptional factor that activates or suppresses the transcription of some genes. In contrast, SCAND1 forms heterodimer with MZF1 and could not bind directly to the DNA, acting as a dominant negative factor that negatively regulates MZF1. If SCAND1 makes a heterodimer with MZF1 to inhibit EMT, does MZF1 promote it? However, these data are not shown. It appears likely that heterodimer between SCAND1 and MZF1 cannot bind to the promoter region of the genes. How does SCAND1 work as an EMT suppressor? Otherwise, can heterodimer between SCAND1 and MZF1 bind directly to the DNA, presented in fig.6A? If so, the data should be shown. Because the molecular mechanisms of SCAND1-inhibiting EMT are not evaluated, it is very hard to understand whole story in this manuscript and the authors’ conclusion. 

Author Response

Thank you for your important comments. Accordingly, we revised the manuscript as follows.

  1. Grammatical errors; 

  line 225, A data set of prostate adenocarcinoma(...) were << was?

  line 229, (CTNNB1) were << was?

  Line 319, E-cadherin were << was? And more

We corrected grammatical errors.

2.The authors’ statements are not correct.

  1) Line162, the pellet was stored.... The sentence should be deleted.

We added new WB data with chromatin fraction (new Fig 2, S1), which were prepared using the pellet. Therefore we edited this sentence as "The insoluble pellet was treated with SDS sample buffer containing β-mercaproethanol, boiled at 95℃ for 10 min, and used as chromatin (Chr.) fractions."

  2) Line261-262, expression levels of MZF1 of Normal cell (left side) is almost similar to those of DU145 cells (2nd from right side) in figure 1D.

Accordingly we corrected the sentence to be "SCAND1 was expressed in normal prostate cells while its levels were reduced in prostate cancer PC-3 and DU-145 cells (Figure 1D), suggesting that SCAND1 expression declines along with prostate oncogenesis. MZF1 was well expressed in epithelial cell line PNT2, although MZF1 was at a lower level in DU-145 and at minimal levels in PC-3 cells as compared to PNT2 cells."

  3) line266-267, among 4 cells used in figure 1D, the authors’ statements are supported by only Pc3 and PNT2 cells, but not other 2 cells. 

Accordingly we edited the sentence to be "These data suggested that SCAND1 expression correlates with epithelial features, whereas the loss of SCAND1 is correlated with mesenchymal features."

  4) line 347, figure 1 clearly indicates “colocalization between SCAND1 and MZF1, but not “direct interaction”. If the authors show it, experiments using recombinant proteins are needed.

We corrected it to be "co-localization".

  1. Cell morphology of normal cells (Figs. 1B, 1C) should be also shown in figure 1A.

We showed a cell morphology of normal cells in figure 1A.

  1. Figure 1F clearly indicated DU-145 cells show 100% infection efficiency. Is it true? Show specificity of both antibodies using siRNAs etc. It remains unclear why anti-SCAND1 antibody is not used in these experiments. In addition, detailed information of anti-MZF1 antibody is missing.

We are sorry for the figure legend was not enough written. The figure was moved to Figure 2F, and we corrected the legend as "(A) Constructs for overexpressing MZF1Flag, SCAND1Flag and SCAND1HA3. These were transfected into DU-145 cells, and stable clones were established." and "(F–H) Confocal microscopy showing co-expression and co-localization between SCAND1Flag vs. MZF1 (F), SCAND1-Flag vs. HP1γ-HA3 (G), or SCAND1 vs. β-catenin. (F) Antibodies against HA-tag and MZF1 were used."

Thus, as written in the method section, we established stable clones overexpressing SCAND1 by the puromycin selection method. Therefore, all the cells (100%) expressed SCAND1 fused with HA3 tag, which was detected using anti-HA tag antibody. Detailed antibody information, including anti-MZF1 antibody, is written in the method section.

  1. What is significance of figure 2B? Rather than figure 2B, expression of both SCAND1 and MZF1 should be shown in panels of figure 2E.

Accordingly we removed the data. To show how SCAND1 and MZF1 were expressed in stable clones, we generated new Figures 2 and S1.

  1. The authors failed to show the cell morphology of MZF1-overexpressed cells. Show it in figure 2C.

Accordingly we added the cell morphology of MZF1-overexpressed cells in the current Figure 3A. Hybrid E/M status was observed in both vector-transfected and MZF1 overexpression DU-145 cells.

  1. E-cadherin in DU-145 cells is almost nothing (Fig 1D), but detected in fig. 2F. In addition, the authors mentioned “E-cadherin faintly detectable in the cytoplasm and possibly in nuclei (line 324-326). Why is it not detected in WB analysis in figure 1D. Since non-specific staining in immunohistochemical study are frequently observed, the authors should exclude this possibility.

Accordingly, we deleted the sentence.

  1. What do the authors want to claim on SCAND1 c1 cells (Fig 2E)? Phenomena in SCAND1 c1 are not completely different from those in other overexpressed cells. Moreover, as mentioned above, expression levels of SCAND1 should be shown.

Accordingly we added the data showing expression levels of SCAND1 in new Fig 2 and Fig S1. As mentioned in the comment, SCAND1 expression level in the c1 clone was lower than c2 and c3. A lower concentration of G418 (2 mg/ml) was used for establishing c1 clone, which could be a cause of the different phenotype, such as a lower anti-EMT effect. We added this interpretation in the result section.

  1. Line 295-296; “the intracellular effects of SCAND1, which lacks a DNA-binding domain, are mediated by binding a SCAN-ZF partner, most likely MZF1 and forming a repressive heterodimer.”

In general, transcription factor that lacks DNA binding domain acts as a dominant negative factor (Curr Top Dev Biol. 2014;110:189-216. doi: 10.1016/B978-0-12-405943-6.00005-1). Thus, it seems that MZF1 forms homodimer to directly bind to the DNA and acts a transcriptional factor that activates or suppresses the transcription of some genes. In contrast, SCAND1 forms heterodimer with MZF1 and could not bind directly to the DNA, acting as a dominant negative factor that negatively regulates MZF1. If SCAND1 makes a heterodimer with MZF1 to inhibit EMT, does MZF1 promote it? However, these data are not shown. It appears likely that heterodimer between SCAND1 and MZF1 cannot bind to the promoter region of the genes. How does SCAND1 work as an EMT suppressor? Otherwise, can heterodimer between SCAND1 and MZF1 bind directly to the DNA, presented in fig.6A? If so, the data should be shown. Because the molecular mechanisms of SCAND1-inhibiting EMT are not evaluated, it is very hard to understand whole story in this manuscript and the authors’ conclusion.

According to the comment, we added the data showing MZF1 positively regulating Vimentin expression (new Fig. 3 J, K). However, we could not conclude whether MZF1 could down- or up-regulate E-cadherin. Because MZF1 contains some isoforms and undergoes several types of PTMs (Figure S1, S2), including phosphorylation, sumoylation and ubiquitination, and interact with PML nuclear bodies, the detailed roles of MZF1 in cancer appear to be complicated (Eguchi T et al 2015 J Cell Biochem). We added these sentences in the discussion section.

Our study also touch upon a mechanism by which SCAND1 represses gene expression involving EMT. Our data indicate that SCAND1 and MZF1 were co-localized to heterochromatin with HP1γ, suggesting that SCAND1-MZF1 complexes involve forming a repressive chromatin status (Figure 2). Consistently, we have shown that overexpressing SCAND1 and MZF1 form oligomers in chromatin and bind to an MZF1-binding site in the promoter region of CDC37 gene, and hetero-oligomers of SCAND1 and MZF1 can repress CDC37 gene expression in prostate cancer [28] (A graphical abstract showing hetero-oligomerization of MZF1 and SCAND1 was published in a supplemental figure). The oligomerizing function of the SCAN domain was first reported by Tucker Collins' group in the paper entitled "the zinc finger-associated SCAN box is a conserved oligomerization domain" [24]. Subsequently, Tara Sander et al. identified a novel SCAN box-related protein called RAZ1 (also known as SCAND1) that interacts with MZF1B to show the leucine-rich SCAN box mediated hetero- and homo-protein associations [26]. Thus, SCAN-ZFs, including MZF1, can mediate repressive role of SCAND1 in gene regulation. In addition to their target gene CDC37, we also have unpublished data demonstrating MZF1 and SCAND1 localized to an MZF1-binding site in the promoter region of HSP90 gene encoding a partner of CDC37. We have shown that molecular chaperones CDC37 and heat shock protein 90 (HSP90) and HSP90-rich exosomes are crucial in promoting EMT in prostate cancer, tongue cancer, and normal epithelial cells [9,10,12]. Therefore, it is speculated that the mechanism by which SCAND1-MZF1 reverse EMT could be transcriptional co-repression of CDC37 and HSP90 genes and a reduction of exosomes. Exact mechanisms should be clarified in the future.

We added these sentences as a paragraph in the discussion section.

We appreciate this reviewer for important points.

Reviewer 3 Report

Thank you for giving me the opportunity to review a very interesting research article. In this manuscript Eguchi and colleagues presented evidence linking EMT with SCAN-zinc finger transcription factors in prostate cancer. Overall, it is a well-written article providing sufficient experimental in vitro and in vivo evidence on the existence of a correlation between SCAND1 and MZF1 with EMT in cancer. The methods are adequately described, and the results are clearly presented. In my opinion, most of the conclusions are supported by the results. Furthermore, the authors could slightly improve the discussion. My few minor comments are listed below:

1. In lines 60-61 the authors mention “Besides, recent studies have shown that EMT is rarely fully executed in tumor cells, while the process is rather gradual and often remains incomplete, termed partial EMT, hybrid EMT or hybrid E/M”. Is the phrase “rarely fully executed in tumor cells” correct or do the authors intended to say that “EMT is rarely executed as an on/off phenomenon in cancer”? Please rephrase.

2. In lines 276-277 the authors mention that “These data suggested that SCAND1-MZF1 co-expression and interaction is involved in maintaining epithelial features”. However, co-expression of SCAND1 and MZF1 does not necessarily suggest that there is interaction. Please explain.

3. In lines 308-309 the authors mention that “Vimentin was markedly reduced by SCAND1 overexpression, although increased by MZF1 over-expression”. Since the authors' hypothesis is that SCAND1/MZF1 suppresses EMT in cancer, could the authors comment on why MZF1 overexpression increases vimentin expression, thus promoting EMT.

4. In lines 325-326 “E-cadherin was faintly detectable in the cytoplasm and possibly in nuclei of non-transfected and MZF1-overexpressed cells”. Since E-cadherin has been widely known as a cell adhesion molecule, this finding of nuclear E-cadherin staining is particularly interesting and may suggest a new biological role. Authors could explain in the discussion the existing data on the significance of nuclear E-cadherin expression.

5. In line 347 the authors claim that “We have shown the coexpression and direct interaction of SCAND1 and MZF1 in Figure 1”. Did the authors provide sufficient evidence on the existence of direct interaction?

6. The authors provided evidence that MZF1 overexpression induces EMT and correlates with poorer prognosis in prostate cancer. Therefore, in the discussion they should mention the existing studies on linking MZF1 with EMT. Is it possible that SCAND1 reverses EMT independently of MZF1 or is it SCAND1-MZF1 complexes that reverse EMT, as the authors hypothesize?

7. It is more appropriate to write “renal cell carcinoma” than “kidney RCC” (tautology).

 My best wishes to the authors.

Author Response

Thank you for your important comments. Accordingly, we revised the manuscript as follows.

  1. In lines 60-61 the authors mention “Besides, recent studies have shown that EMT is rarely fully executed in tumor cells, while the process is rather gradual and often remains incomplete, termed partial EMT, hybrid EMT or hybrid E/M”. Is the phrase “rarely fully executed in tumor cells” correct or do the authors intended to say that “EMT is rarely executed as an on/off phenomenon in cancer”? Please rephrase.

Accordingly we rephrased it.

  1. In lines 276-277 the authors mention that “These data suggested that SCAND1-MZF1 co-expression and interaction is involved in maintaining epithelial features”. However, co-expression of SCAND1 and MZF1 does not necessarily suggest that there is interaction. Please explain.

We excluded "and interaction" from this sentence.

  1. In lines 308-309 the authors mention that “Vimentin was markedly reduced by SCAND1 overexpression, although increased by MZF1 over-expression”. Since the authors' hypothesis is that SCAND1/MZF1 suppresses EMT in cancer, could the authors comment on why MZF1 overexpression increases vimentin expression, thus promoting EMT.

Accordingly we added data showing MZF1 positively regulating Vimentin gene expression, a marker of mesenchymal cells  (Fig. 3 C, D). Overexpression of MZF1 can increase the number of MZF1 homo-oligomer (dimer or trimer) that could activate transcription and EMT. However, we additionally found that MZF1 and SCAND1 are mutually inducible (new Figure 2). Co-expression and hetero-oligomerization of SCAND1 and MZF1 is a repressor of transcription and EMT.

  1. In lines 325-326 “E-cadherin was faintly detectable in the cytoplasm and possibly in nuclei of non-transfected and MZF1-overexpressed cells”. Since E-cadherin has been widely known as a cell adhesion molecule, this finding of nuclear E-cadherin staining is particularly interesting and may suggest a new biological role. Authors could explain in the discussion the existing data on the significance of nuclear E-cadherin expression.

Accordingly we added a figure to further examine nuclear E-cadherin (Figure 3 M, N) and discussion:

Our study also suggested that nuclear E-cadherin could be involved in reverse EMT. We showed nuclear E-cadherin in the SCAND1-overexpressing cells with reverse EMT status (Figure 3). Ferber et al., for the first time, reported a role for the cleaved cytoplasmic domain of E-cadherin in the nucleus [56]. Subsequently, Cespedes et al. reported site-dependent E-cadherin cleavage and nuclear translocation in a metastatic colorectal cancer model [57]. Su et al. reported that an aberrant nuclear localization of E-cadherin is a potent inhibitor of Wnt/β-catenin-elicited promotion of the cancer stem cell phenotype [58]. On the other hand, Zhao et al. reported that nuclear E-cadherin acetylation promoted colorectal tumorigenesis via enhancing β-catenin activity. In our current study, nuclear E-cadherin was seen in the plasmid-based SCAND1 overexpressing cells, although less found in lentivirus-mediated SCAND1 overexpressing cells. Nevertheless, it will be of great importance in the future how nuclear E-cadherin and SCAND1 are involved in regulating EMT and cancer stem cells.

  1. In line 347 the authors claim that “We have shown the co-expression and direct interaction of SCAND1 and MZF1 in Figure 1”. Did the authors provide sufficient evidence on the existence of direct interaction?

We corrected the sentence as "We have reported that MZF1 and SCAND1 formed oligomers in chromatin and associated to "an MZF1-binding site" in the promoter region of CDC37 gene, indicating MZF1/SCAND1 hetero-oligomers as transcriptional repressor complex {Eguchi et al, 2019 Cancers}. We also have shown that SCAND1 and MZF1 were co-expressed and co-localized in nuclei in prostate cancer cells (Figure 2)."

  1. The authors provided evidence that MZF1 overexpression induces EMT and correlates with poorer prognosis in prostate cancer. Therefore, in the discussion they should mention the existing studies on linking MZF1 with EMT. Is it possible that SCAND1 reverses EMT independently of MZF1 or is it SCAND1-MZF1 complexes that reverse EMT, as the authors hypothesize?

Accordingly we added data showing MZF1 positively regulating Vimentin gene expression, a marker of mesenchymal cells  (Fig. 3 J, K), which could be a cause of poor prognosis of patients with MZF1 high prostate adenocarcinomas. Consistently, we have shown that MZF1-targeted siRNA inhibited prostate tumor growth in mouse tumor xenograft model (Eguchi T et al 2019 Cancers). Overexpression of MZF1 can increase the number of MZF1 homo-oligomer (dimer or trimer) activating transcription and EMT. On the other hand, mutually induced co-expression (Fig 2) and hetero-oligomerization of SCAND1 and MZF1 function as a repressor of transcription and EMT.

  1. It is more appropriate to write “renal cell carcinoma” than “kidney RCC” (tautology).

We corrected it.

We appreciate this reviewer for important comments.

Reviewer 4 Report

The manuscript entitled “SCAND1 Reverts EMT and Suppresses Tumor Growth and Collective Migration” by Eguchi et al. is a nice addition to the field of EMT/MET.  The manuscript is generally well written (there are a few grammatical errors that likely will be addressed by the copy editor). Their overall conclusion that SCAND1 reverses EMT is a bit strong given some of the data they show, variation in clones, and the lack of statistical analysis of much of the data. Their claim that co-expression =  colocalization, plus their claim that co-expression = co-regulation/co-repression is suggestive but not fully supported by their data. If these data were shown in a previous manuscript (I suppose not as references to Figures is made in the discussion), this should be more clearly stated in the manuscript.  I do have some more substantial comments that would strength the manuscript as well as some minor comments that hopefully can be easily addressed.

Substantial comments:

1.       Figure 1A: are these stable appearances of the cells in culture?  In other words, are PC-3 cells consistently mesenchymal, DU-145 cells consistently a mixed population, etc (are stable)?  Does their morphology depend on how long cells are in culture and/or how confluent the cells are?  A more thorough description in the text (lines 244-252) would be nice to see.  Additionally, is there a reason the authors did not include normal (PrEC) prostate cells for a comparison in this panel?

2.       Along with my previous comment, how long were cells in culture prior to qPCR analysis of ECAD and VIM?  For WB analysis?  Was a consistent time used and/or a consistent confluency of cells used?

3.       Figure 1C: Vimentin expression is lost for PNT2 cells (described on line 257) implies this is a significant reduction. If so, stats should be included. If it is not significant compared to PrEC (normal cells), then why describe this as lost compared to normal cells?  

4.       Is the difference in vimentin expression significant in the WB (Fig 1D)?  Statistical analysis should be done on WB expression in Fig 1D.  Also, on line 261, the authors state that MZF1 is well expressed in both PTN2 and normal, and reduced in DU-145 and significantly reduced in PC-3.  I see that MZF1 expression looks the same in normal and DU-145 (again, no stats).  Their data do not fully support their claim that MZF1 expression correlates with epithelial features, especially since they don’t show PrEC morphology in Figure 1A.

5.       Figure 1F: I agree that SCAND1 and MZF1 are both found within the nucleus, but how are the authors concluding that they are interacting with each other?  Was any colocalization analysis performed?  How are the authors able to conclude (lines 272-275) that they are co-expressed (regulated together) or interacting with each other based on this analysis/these data shown?

6.       The authors make two important assumptions on lines 294-298.  What is the basis for these assumptions?  Some more data/explanation is needed to justify these assumptions.

7.       Figure 2E:  There are three clones for SCAND1 represented (as the text indicates three clones were made for each, line 138) but why is there only one clone shown for MZF1?  Did they not vary? What is the significance of the difference in protein expression in the three SCAND1 clones? I also don’t agree with the authors conclusions that overexpression with MZF1 increased both epithelial and mesenchymal markers. There is no data analysis performed, and with the varying results for the three SCAND1 clones, this is hard to interpret.  Clone SCAND1c1 shows similar ECAD expression as MZF1. The authors need to discuss the variation in clones c1-c3 and provide statistics to support claims made in the text (lines 307-312).  Same comments for other protein expression results in this figure.

8.       Figure 2F: the authors claim “markedly reduced” expression of beta-catenin and relocation, but these figures have not been statistically analyzed, nor do they represent a clear depiction of what the authors are stating. 

9.       Lines 367-372: How do their data support the claims made?  If this is speculation, it should be moved to the discussion.

10.   Figure 4B: The authors claim that reduced vimentin expression supports EMT reversal (lines 395-396). Can the authors make this claim based on IHC data?  The morphology of the cells, at the magnification shown, looks identical, which contrasts with earlier data. 

 Minor comments:

1.       Section 2.2:  Did the authors validate their primers prior to using them?  I would have liked to see a statement that this was done to ensure optimal quality and amplification as they used self-designed primers.

2.       Figure 1E: the authors state there is a significant correlation, but do not show the regression analysis/statistics.  Please include these.

3.       Figure 2A: The description of the expression vectors is unclear; I’m not sure this figure is needed. I am unable to interpret it as presented.

4.       Line 363: the authors state that SCAND1 and MZF1 co-expression is negatively correlated with TGFBRs – this seems incorrect (they aren’t showing co-expression?).

5.       Figure 4: Can the authors show better images, and provide analysis, of colocalization?  I believe some co-expression is occurring, as the authors indicate, but they don’t prove colocalization; it seems as if the authors are using these two terms interchangeably.   Figure 1F showed SCAND1 expression in the nucleus – Figure 4A is at a very low magnification and hard to see this.

6.       Is there a stray vertical line in Figure 5, panel for NHSCC with high MZF1 expression?  Or does survival go from 80% to 0%?  The bottom two panels use blue lines for low expression, which doesn’t match the other panels/key (which are black lines).

7.       The discussion mentions regulation of HSPs and chaperones and the link between SCAND1 and MZF1 expression (Lines 498-513) citing previous data published. A bit more discussion here - putting in a summary of the findings that support their claim, would be appreciated. Otherwise, this discussion seems out of place for the data presented in the results section.

8.       I would have liked to see the discussion of the clinical data (lines 514-525) tied into their discussion on cancer therapy (starting on line 474). The authors have interesting data to conclude that not all therapies targeting SCAND1 and MZF1 (or their signaling pathways) would be equally effective, or therapeutically useful – in fact, for kidney RCC, it would be harmful!  This would be interesting to discuss/tie together in the discussion.

Author Response

Thank you for your important comments. Accordingly, we revised the manuscript as follows.

  1. Figure 1A: are these stable appearances of the cells in culture? In other words, are PC-3 cells consistently mesenchymal, DU-145 cells consistently a mixed population, etc (are stable)? Does their morphology depend on how long cells are in culture and/or how confluent the cells are? A more thorough description in the text (lines 244-252) would be nice to see. Additionally, is there a reason the authors did not include normal (PrEC) prostate cells for a comparison in this panel?

Accordingly we edited the part as:

It has been kown that prostate cancer cell line PC-3 is the most malignant, metastatic, castration-resistant prostate cancer cell line, while DU-145 is a moderately metastatic, prostate adenocarcinoma cell line [13,28,46]. We have experienced that both Vimentin and E-cadherin were detectable in PC-3 cells under several culture conditions, suggesting that prostate cancer cells could be often E/M hybrid status [9,13]. We also used prostate normal epithelial cell line PNT2 and first examined the morphologies of PC-3, DU-145 and PNT2 in the monolayer culture. PC-3 and DU-145 cells were mixed populations of spindle-shaped, mesenchymal cells and the remainder with the cobblestone morphology, a characteristic of epithelial cells, indicating that DU-145 and PC-3 were a hybrid E/M status (Figure 1A). In contrast, PNT2 cell populations contained only epithelial cells with the cobblestone morphology.

We added a picture of normal prostate cells in Fig 1A.

  1. Along with my previous comment, how long were cells in culture prior to qPCR analysis of ECAD and VIM? For WB analysis? Was a consistent time used and/or a consistent confluency of cells used?

To compare different cell types as fair as we could, we seeded cells to be 10% confluent in each dish. Media were changed one day before the sampling for all cell types. We corrected total RNA and proteins at around 50% confluent status of all the cell types tested. We added these sentences in the method section.

We also added a sentence as "E-cadherin levels were consistent with their mRNA expression. (Note that E-cadherin appeared to be lost in PC-3 and DU-145 in this relative western blotting analysis comparing with completely epithelial PNT2 cells, although E-cadherin could be detectable in PC-3 in different culture conditions as we have shown {Eguchi, 2020 Cells;Eguchi, 2018 PLOS One}).

The "relative" difference shown in figures could be caused by differences between cell types and growth factors specified for culturing each cell type. In the WB analysis, E-cadherin appeared to be lost in PC-3 and DU-145. However, this result was caused by the relative WB analysis compared with completely epithelial cells. We believe longer exposure time in WB or different culture conditions could bring E-cadherin detectable in these cancer cells. Indeed, as added in the text, we have detected E-cad and Vimentin in PC-3 in previous studies (Eguchi T et al 2018 PLOS One).

  1. Figure 1C: Vimentin expression is lost for PNT2 cells (described on line 257) implies this is a significant reduction. If so, stats should be included. If it is not significant compared to PrEC (normal cells), then why describe this as lost compared to normal cells?

We added statistics there.

  1. Is the difference in vimentin expression significant in the WB (Fig 1D)? Statistical analysis should be done on WB expression in Fig 1D. Also, on line 261, the authors state that MZF1 is well expressed in both PTN2 and normPal, and reduced in DU-145 and significantly reduced in PC-3. I see that MZF1 expression looks the same in normal and DU-145 (again, no stats). Their data do not fully support their claim that MZF1 expression correlates with epithelial features, especially since they don’t show PrEC morphology in Figure 1A.

We added quantification and statistics of WB analysis (Figure 3 F-I).

We corrected the description of Vimentin as "Vimentin was highly expressed in DU-145 cells and well detectable in PC-3 and normal prostate cells, although not detectable in PNT2 cells."

Regarding MZF1, we corrected the sentences as "MZF1 was well expressed in epithelial cell line PNT2, although MZF1 was at a lower level in DU-145 and at minimal levels in PC-3 cells as compared to PNT2 cells."

  1. Figure 1F: I agree that SCAND1 and MZF1 are both found within the nucleus, but how are the authors concluding that they are interacting with each other? Was any colocalization analysis performed? How are the authors able to conclude (lines 272-275) that they are co-expressed (regulated together) or interacting with each other based on this analysis/these data shown?

They are co-expressed and co-localized in nuclei, and we removed the word "interaction".

  1. The authors make two important assumptions on lines 294-298. What is the basis for these assumptions? Some more data/explanation is needed to justify these assumptions.

We removed these sentences.

  1. Figure 2E: There are three clones for SCAND1 represented (as the text indicates three clones were made for each, line 138) but why is there only one clone shown for MZF1? Did they not vary? What is the significance of the difference in protein expression in the three SCAND1 clones? I also don’t agree with the authors conclusions that overexpression with MZF1 increased both epithelial and mesenchymal markers. There is no data analysis performed, and with the varying results for the three SCAND1 clones, this is hard to interpret. Clone SCAND1c1 shows similar ECAD expression as MZF1. The authors need to discuss the variation in clones c1-c3 and provide statistics to support claims made in the text (lines 307-312). Same comments for other protein expression results in this figure.

Accordingly we added data showing MZF1 clones in new Fig 2. They varied, and we chose MZF1 clone 2, which expressed MZF1 at the highest level.

In the same figure, we showed different SCAND1 expressions in SCAND1 expressing clones. SCAND1 expression levels were higher in c2 and c3 than c1, which could cause different expressions of E-cad and p-ERK between these clones (Fig 2, S1, S2).

We also added data showing MZF1 positively regulates VIM expression. However, we do not have a conclusion on how MZF1 regulates ECAD. Because MZF1 has isoforms and underwent PTMs (Figure S1), it could be difficult to determine the exact roles of MZF1 in EMT.

Regarding the variation of SCAND1 clones c1-c3, we added the detailed data in new Fig 2, S1, and S2 and interpretation as;

Overexpressing SCAND1 was bound to chromatin in the SCAND1 c1, c2 and c3 clones (Figure 2I, top panel; Figure 2J, S1, S2). Soluble SCAND1 was lower in the SCAND1 c1 clone compared to c2 and c3 clones. (The difference might be caused by a lower concentration of G418 used for establishing the c1 clone.)

We provided statistics in new Fig 3 F-I.

  1. Figure 2F: the authors claim “markedly reduced” expression of beta-catenin and relocation, but these figures have not been statistically analyzed, nor do they represent a clear depiction of what the authors are stating.

Accordingly, we corrected the sentence: "SCAND1 overexpression appeared to reduce nuclear β-catenin and led to relocation to intercellular adhesion sites formed by E-cadherin.

  1. Lines 367-372: How do their data support the claims made? If this is speculation, it should be moved to the discussion.

Accordingly we removed these sentences.

  1. Figure 4B: The authors claim that reduced vimentin expression supports EMT reversal (lines 395-396). Can the authors make this claim based on IHC data? The morphology of the cells, at the magnification shown, looks identical, which contrasts with earlier data.

Accordingly we added new Fig 5C with high magnification to show that vimentin-positive cells were spindle-shaped, whereas vimentin-negative cells were epithelial morphology.

 Minor comments:

  1. Section 2.2: Did the authors validate their primers prior to using them? I would have liked to see a statement that this was done to ensure optimal quality and amplification as they used self-designed primers.

We added, "We validated the primer pairs by melting curve analysis of single amplicons, amplification efficiencies, band size in agarose gel electrophoresis, and DNA sequencing of PCR products."

  1. Figure 1E: the authors state there is a significant correlation, but do not show the regression analysis/statistics. Please include these.

We added Spearman's and Pearson's correlation scores with p values and regression analysis in Figure 1E.

  1. Figure 2A: The description of the expression vectors is unclear; I’m not sure this figure is needed. I am unable to interpret it as presented.

We revised the structures and names of the constructs.

  1. Line 363: the authors state that SCAND1 and MZF1 co-expression is negatively correlated with TGFBRs – this seems incorrect (they aren’t showing co-expression?).

We rephrased it as "SCAND1 and MZF1 'expression' was also negatively correlated with TGFBR1, TGFBR2 and TGFBR3 gene expression".

  1. Figure 4: Can the authors show better images, and provide analysis, of colocalization? I believe some co-expression is occurring, as the authors indicate, but they don’t prove colocalization; it seems as if the authors are using these two terms interchangeably. Figure 1F showed SCAND1 expression in the nucleus – Figure 4A is at a very low magnification and hard to see this.

Accordingly we showed better images with higher resolutions and DAPI staining (current Fig 5).

As pointed out, we corrected the description as "Enhanced co-expression of E-cadherin was seen in/around the cells overexpressing SCAND1".

  1. Is there a stray vertical line in Figure 5, panel for NHSCC with high MZF1 expression? Or does survival go from 80% to 0%? The bottom two panels use blue lines for low expression, which doesn’t match the other panels/key (which are black lines).

We do not think the survival rate suddenly went down from 80% to 0%. It could be an error in the software. Therefore, we removed the stray vertical lines.

For the upper 3 cancer types, we analyzed data using KM plotter portal, which provided black lines for low-expression groups. However, prostate AC data were not available in the KM plotter. Therefore, for prostate AC, we used GEPIA2 portal, which provided blue lines. We corrected the color of lines as better as we could and believe the reader could understand the figure.

  1. The discussion mentions regulation of HSPs and chaperones and the link between SCAND1 and MZF1 expression (Lines 498-513) citing previous data published. A bit more discussion here - putting in a summary of the findings that support their claim, would be appreciated. Otherwise, this discussion seems out of place for the data presented in the results section.

We edited the part to be;

We have shown that molecular chaperones CDC37 and heat shock protein 90 (HSP90) and HSP90-rich exosomes are crucial in promoting EMT in prostate cancer, tongue cancer, and normal epithelial cells [9,10,12]. Therefore, it is speculated that the mechanism by which SCAND1-MZF1 reverse EMT could be transcriptional co-repression of CDC37 and HSP90 genes and a reduction of exosomes. Exact mechanisms should be clarified in the future.

Our data suggest that SCAND1 and MZF1 might also regulate molecular chaperone expression and the cell stress response in cancer. Cell stress, such as heat shock stress, activates heat shock transcription factor 1 (HSF1) that induces stress-resopnsive chaperone expression including HSP90 and CDC37 [9]. Subsequently the molecular chaperone CDC37 is an essential regulator for exosome production [9]. Therefore, it is suggested that tanscription factors HSF1, MZF1, and SCAND1 co-regulate cell stress response, molecular chaperone gene expression, and exosome production. Detailed mechanisms should be clarified in the future.

  1. I would have liked to see the discussion of the clinical data (lines 514-525) tied into their discussion on cancer therapy (starting on line 474). The authors have interesting data to conclude that not all therapies targeting SCAND1 and MZF1 (or their signaling pathways) would be equally effective, or therapeutically useful – in fact, for kidney RCC, it would be harmful! This would be interesting to discuss/tie together in the discussion.

Accordingly we moved the prognostic part of the discussion to attach to the therapeutic strategy paragraph. Because these paragraphs are closely related, as commented, it would be effective to increase the readability and understandability.

We appreciate this reviewer for important comments.

Round 2

Reviewer 4 Report

The authors have addressed each of my comments and I thank them for the detailed explanations they have provided. I realize that they added in a new Figure 2 in order to establish the colocalization of these proteins, which was nice to see included.  There have been significant improvements to the manuscript, but I still have a few comments that should be addressed.  Also, grammatical and tense errors still exist within sentence structure – I did not point out specific examples last time but should be addressed/corrected.

1.  Figure 1A:  I don’t see that PTN2 cells only contain cells with an epithelial cell morphology.  While it is not as mixed as PC-3 and DU-145, PTN2 cells look quite different, to me, from the normal epithelial cells. This should be addressed in some fashion.

2.  I don’t understand this sentence in the author’s comments to my comments about consistency of sampling for qPCR:  “We corrected total RNA and proteins at around 50% confluent status of all the cell types tested.”  On lines 135-137 of the manuscript, they state: “To compare the E/M status of different cell types, we first seeded cells to be 10% confluent in each dish. We replaced media with fresh ones one day before the sampling, then corrected samples at around 50% confluent status of cells.”  What does it mean to “correct samples”?   On lines 184-185, the authors state “Total RNA was collected after 48 hours.” Was this also at 50% confluency or didn’t matter how many cells were in culture after 48 hours. No mention of correcting for amount of RNA in analyses in either section (2.4 or 2.5).

3.  Figure 1B and 1C: The authors did multiple t-tests on the data (it appears from the description in section 2.12) yet there was no mention of performing any multiple-comparison corrections, such as a Bonferroni correction.  Lines 277-279 mention all stats were done comparing two groups yet some figures did multiple comparisons.

4.  Figure 3 seems to be missing in the version of the manuscript I downloaded, so I cannot evaluate the data or interpretation of results.

5.  A minor point, but the authors changed “reverts” to “reverses” in the title, but still use “reverts” in places throughout the manuscript (i.e., lines 30, 125, 586 and 641).  

Author Response

Thank you for your important comments. Accordingly, we revised the manuscript as follows.

We corrected grammatical and tense errors and highlight in yellow.

  1. Figure 1A:  I don’t see that PTN2 cells only contain cells with an epithelial cell morphology.  While it is not as mixed as PC-3 and DU-145, PTN2 cells look quite different, to me, from the normal epithelial cells. This should be addressed in some fashion.

We added black arrowheads indicating spindle-shaped cells in the pictures of normal cells and PNT2 in Fig 1A. The sentence was revised: Prostate normal cells and PNT2 cell populations mostly appeared as epithelial cells with cobblestone morphology and intercellular adhesion and rarely contained spindle-shaped cells (Figure 1A).

Prostate-derived normal cells might contain fibroblasts while PNT2 grew as colonies keeping intercellular adhesion, suggesting that PNT2 might be more homogenous epithelial cells than the normal cells.

  1. I don’t understand this sentence in the author’s comments to my comments about consistency of sampling for qPCR:  “We corrected total RNA and proteins at around 50% confluent status of all the cell types tested.”  On lines 135-137 of the manuscript, they state: “To compare the E/M status of different cell types, we first seeded cells to be 10% confluent in each dish. We replaced media with fresh ones one day before the sampling, then corrected samples at around 50% confluent status of cells.”  What does it mean to “correct samples”?   On lines 184-185, the authors state “Total RNA was collected after 48 hours.” Was this also at 50% confluency or didn’t matter how many cells were in culture after 48 hours? No mention of correcting for amount of RNA in analyses in either section (2.4 or 2.5).

We revised the sentences in section 2.1: We replaced media with fresh ones one day before taking pictures and sampling proteins and RNA, then corrected samples at around 50% confluent status of cells.

Regarding siRNA transfection written in section 2.4, because this is siRNA transfection followed by qRT-PCR, transfection and knockdown efficiencies have a higher priority. As a regular protocol also recommended by the manufacturer, we used the reverse-transfection method, which resulted in transfection at 40 – 60% confluent status.

Total RNA is usually collected at 48 h post-transfection period, often at 80 – 90% confluent status, as we did this time as well. Again, this is a regular, established protocol prioritizing transfection and knockdown efficiencies, and we confirmed an efficient knockdown of MZF1 using this protocol (Eguchi T et al. 2019 Cancers).

  1. Figure 1B and 1C: The authors did multiple t-tests on the data (it appears from the description in section 2.12) yet there was no mention of performing any multiple-comparison corrections, such as a Bonferroni correction.  Lines 277-279 mention all stats were done comparing two groups yet some figures did multiple comparisons.

We did not perform multiple-comparison corrections. To avoid confusion and misunderstanding, we revised Figure 1 B and C. As written in section 2.12. statistics, we compared values of two groups. For example, in Fig 1B, we compared values of Normal vs. PC-3, then Normal vs. DU-145.

According to Wikipedia, "Bonferroni correction is a conservative method that gives greater risk of failure to reject a false null hypothesis than other methods, as it ignores potentially valuable information, such as the distribution of p-values across all comparisons (which, if the null hypothesis is correct for all comparisons, is expected to take a uniform distribution)."  https://en.wikipedia.org/wiki/Bonferroni_correction                                                                     

  1. Figure 3 seems to be missing in the version of the manuscript I downloaded, so I cannot evaluate the data or interpretation of results.

The uploaded version included Figure 3 which was confirmed by the assistant editor and other reviewers. We hope you could find Figure 3 this time. 

  1. A minor point, but the authors changed “reverts” to “reverses” in the title, but still use “reverts” in places throughout the manuscript (i.e., lines 30, 125, 586 and 641).  

We changed revert to reverse throughout.

We appreciate the comments this reviewer who brings up important points.